# Development of Antioxidant and Antimicrobial Membranes Based on Functionalized and Crosslinked Chitosan for Tissue Regeneration

**DOI:** 10.3390/ijms25041961

**Published:** 2024-02-06

**Authors:** Clarissa Ciarlantini, Elisabetta Lacolla, Iolanda Francolini, Marta Fernández-García, Carolina Muñoz-Núñez, Alexandra Muñoz-Bonilla, Antonella Piozzi

**Affiliations:** 1Department of Chemistry, Sapienza University of Rome, Piazzale A. Moro, 5, 00185 Rome, Italy; clarissa.ciarlantini@uniroma1.it (C.C.); elisabetta.lacolla@uniroma1.it (E.L.); iolanda.francolini@uniroma1.it (I.F.); 2Institute of Polymer Science and Technology (ICTP-CSIC), Juan de la Cierva 3, 28006 Madrid, Spain; martafg@ictp.csic.es (M.F.-G.); carolinamunoz@ictp.csic.es (C.M.-N.); sbonilla@ictp.csic.es (A.M.-B.); 3Interdisciplinary Platform for Sustainable Plastics towards a Circular Economy-Spanish National Research Council (SusPlast-CSIC), 28006 Madrid, Spain

**Keywords:** antimicrobial membranes, chitosan membrane, 3,4 hydroxycinnamic acid, adhesive membranes, wound healing

## Abstract

Tissue engineering is an interdisciplinary field that develops new methods to enhance the regeneration of damaged tissues, including those of wounds. Polymer systems containing bioactive molecules can play an important role in accelerating tissue regeneration, mitigating inflammation process, and fighting bacterial infection. Chitosan (CS) has attracted much attention regarding its use in wound healing system fabrication thanks to its biocompatibility, biodegradability, and the presence of functional groups in its structure. In this work, bioactive chitosan-based membranes were obtained by both chemical and physical modifications of the polymer with glycidyl methacrylate and glycerol (GLY), respectively. The most suitable GLY concentration to obtain wound healing systems with good elongation at break, a good water vapor transmission rate (WVTR), and good wettability values was 20% (*w*/*w*). Afterwards, the membranes were crosslinked with different concentrations of ethylene glycol dimethacrylate (EGDMA). By using a concentration of 0.05 mM EGDMA, membranes with a contact angle and WVTR values suitable for the application were obtained. To make the system bioactive, 3,4-dihydrocinnamic acid (HCAF) was introduced into the membranes, either by imbibition or chemical reaction, using laccase as a catalyst. Thermal and mechanical analyses confirmed the formation of a cohesive network, which limited the plasticizing effect of GLY, particularly when HCAF was chemically bound. The HCAF-imbibed membrane showed a good antioxidant and antimicrobial activity, highlighting the potential of this system for the treatment of wound healing.

## 1. Introduction

Currently, there are numerous opportunities and challenges in the development and characterization of biomaterials. In the medical field, the functions required for biomaterials are multiple. They can be exploited, for example, to improve joint capacity, to control blood flow, in the filling of cavities (in the context of cosmetic surgery), and for the distribution of drugs or other bioactive molecules and are widely used in the regeneration of tissues [1,2]. To allow for the realization of all these applications, biomaterials can be modelled in different geometries. For example, in tissue engineering (TE) applications, three-dimensional porous architectures, used as supports (scaffolds), as well as flat structures (film) for wound healing, can be manufactured. 

Wound healing is a complex biological process related to the regeneration of damaged tissue, which is divided into the following phases: hemostasis, inflammation, fibroblast migration, proliferation, and the formation of connective tissue [3]. Inflammatory and infectious phenomena can significantly slow this process down. Inflammation is a natural process during wound healing and is essential for the removal of microorganisms [4]. However, in the absence of effective decontamination, the inflammatory process can be prolonged with an increase in the concentration of pro-inflammatory cytokines [5]. This phenomenon has the potential to induce chronicity in the wound, causing the healing process to fail. In addition, all wounds have a high microbial colonization. Wound infection is, therefore, highly detrimental to wound healing. The possibility that the microbial colonization of a wound evolves into an infection depends on the patient’s immune system and, above all, on the types of microorganisms that are involved and their virulence factors [6]. A critical virulence factor is the organism’s ability to form microbial biofilm [7]. Bacteria in biofilms have an altered phenotype compared to the planktonic form, in terms of growth rate and gene transcription, and thus acquire protection against host defenses and antimicrobial agents [8]. Generally, dressings are used to promote the healing process of wounds. An ideal dressing should be impermeable to water and bacteria, allow for good gas exchange, absorb excess exudate, maintain adequate moisture in the wound bed, and provide thermal insulation to the wound [9,10,11]. Wound dressings can be classified as inert, interactive, or bioactive dressings. Bioactive dressings can function as a barrier against bacterial infections, modify the physiology of the wound environment, improve re-epithelialization, provide a moist wound environment, and, importantly, deliver bioactive agents that are functional in the healing process [12]. Therefore, in recent years, research has concentrated on designing polymer systems that are able to release molecules (antioxidants, anti-inflammatories, or drugs), thus avoiding harmful processes for the body. Endogenous and exogenous stimuli–response systems based on nanomaterials appear to be very promising for biomedical applications including wound healing, as they not only release the drug in a controlled manner at the target site but can also contribute to counteracting the problem of multi-drug-resistant bacterial wound infection [13,14]. 

In the fabrication of dressings, natural polymers are gaining more and more interest [15]. Polysaccharides such as hyaluronic acid (HA), alginate (AL), and chitosan (CS) have attracted much attention for wound dressing fabrication thanks to their excellent properties, such as their biocompatibility (similarity to the extracellular matrix (ECM)), biodegradability, non-toxicity, and easy processability. Furthermore, the presence of functional groups in their chains makes them easily modifiable, both through chemical reactions and physical interactions with different molecules, allowing for the development of materials with better antibacterial and/or anti-inflammatory activity. CS is one of the most used, thanks to its intrinsic antimicrobial properties deriving from the presence of the amino group in the chain [16,17], and its ability to promote the formation of granulation tissue [18,19]. Indeed, as reported by P. Shivakumar et al., from 2010 to 2020, several patents on the application of CS and its derivatives in wound healing were developed, confirming the high applicability of this polymer [20]. CS’ ability to limit bleeding phenomena has also allowed for the production of commercial dressings such as ChitoGauze XR pro or HemCon GuardaCare^®^ pro. However, CS is characterized by poor dimensional stability in an aqueous environment and mechanical properties that are not suitable for wound healing applications. For these reasons, it needs to be chemically or physically modified. 

Generally, CS in contact with aqueous systems and physiological environments tends to absorb large quantities of the aqueous phase, which leads to a non-reversible structural modification with an associated loss of physical properties. For this reason, the cross-linking of the polymer is of fundamental importance in the creation of systems to be used in TE. CS can be crosslinked using physical crosslinkers, exploiting the electrostatic interactions between groups of opposite charge and/or hydrogen bonds [21,22], or using chemical ones that are capable of forming covalent bonds that stabilize the structure. In general, chemical cross-linking is preferred in the production of dressings, as it leads to more stable structures that are able to carry out their activity when in contact with the physiological environment of the wound for a longer period of time. The generally used chemical agents are genipin, glutaraldehyde, poly(ethylene glycol diglycidyl ether) (PEGDE), and epichlorohydrin [23,24,25,26]. However, some strategies include the possibility of CS interacting with polymers containing carboxyl groups (alginate or hyaluronic acid or other modified polymers) to obtain the formation of polyelectrolyte complexes that are subsequently stabilized with cross-linking agents [27,28] or to exploit EDC:NHS reagents to form amide bonds, as reported by Zhang et al. [29]. The cross-linking process can also be used to improve the bioactive dressing’s ability to release or encapsulate drugs or molecules of interest in the healing process. In fact, by varying the ratio between the polymer and cross-linking agent, it is possible to modify the release kinetics of the loaded bioactive molecules, allowing for the creation of versatile structures that are applicable in clinical practice [30].

However, in dressing’s production through the use of the solvent casting technique, CS films are characterized by their high tensile strength, which results in a low flexibility and high fragility [31]. Furthermore, the subsequent chemical cross-linking process also tends to worsen this behavior, producing more stable but also more fragile matrices, characterized by severe deformations such as shrinkage and curling. For these reasons, generally, in the production of CS-based dressings, it is necessary to introduce plasticizing agents, such as polyols (glycerol (GLY), sorbitol, and poly(ethylene glycol)) or fatty acids (stearic and palmitic acids), that help to overcome CS’ brittleness [32,33]. The introduction of these molecules among CS chains allows for greater flexibility, also modifying the ability of the dressing to interact with the physiological environment, thanks to the formation of a network of H bonds [34,35]. Of all the plasticizers, glycerol is one of the most used, due to its high availability, lack of toxicity, and low exudation. In addition, as reported by Epure et al. [36], the introduction of GLY in CS-based dressings makes it possible to obtain better mechanical properties but, above all, greater flexibility, which is essential for obtaining an adhesive and long-lasting dressing.

As previously mentioned, CS dressings are characterized by good antimicrobial properties. However, it has been demonstrated that, during the inflammatory phase of the wound, there is an overproduction of reactive oxygen species (ROS) by activated phagocytes [37]. High levels of ROS are responsible for poor wound healing [38] and promote antibiotic resistance due to their accumulation in microbial biofilms [39]. To reduce the formation of ROS during the inflammation stage, the introduction of antioxidant molecules in the design of dressing can be a good strategy [40]. In fact, as reported by Abbas et al., the introduction of an antioxidant molecule, such as curcumin, can improve the antioxidant properties of CS, which are generally not sufficient for ROS capture [41]. Furthermore, the introduction of some classes of antioxidant molecules, such as catechols, can improve the adhesion of the dressing to damaged tissues, promoting the regeneration of the latter [42]. Indeed, in the development of biomedical adhesive surfaces, the adhesion phenomena of mussels were taken as an example. Thanks to the presence of 3,4-dihydroxy-L-phenyalanine (DOPA), a catecholic amino acid present in the adhesive proteins of mussels, mussels can penetrate the boundary layers of water and form bonds with different substrates [43]. This has led to the development of polymeric systems functionalized with molecules containing catechol groups [44]. As reported by J. H. Ryu et al., 3,4 hydroxycinnamic acid-functionalized CS and pluronic composite hydrogels showed strong adhesiveness to soft tissues and mucous layers, and good haemostatic properties compared to the same catechol-free composite hydrogels [45]. However, in developing the dressing, the method of introduction of the antioxidant molecule must also be considered. The introduction can take place chemically through a covalent bond with the matrix or through imbibing processes, which exploit the simple ionic interactions between the matrix and the molecule of interest. In the case of covalent bond formation, there is the advantage of the greater degree of encapsulation of the bioactive molecule. However, unlike the imbibition process, the molecule could be less reactive, given its low mobility and the possibility that, with the formation of the covalent bond, fundamental functions for the capture of radicals are undermined.

In this work, bioactive membranes from CS with potential applications in the regeneration of damaged skin tissues were developed by the solvent casting technique. CS, chosen due to its biological properties, which are suitable for producing systems for TE applications, was initially modified with glycidyl methacrylate (GMA) and glycerol (GLY), with this latter used as a plasticizer, and then crosslinked with the difunctional monomer, ethylene glycol dimethacrylate (EGDMA), to increase the developed membrane stability in a humid environment. Furthermore, to avoid possible inflammatory reactions during the skin reparation phase and improve the adhesive and antimicrobial properties of the obtained devices, 3,4 hydroxycinnamic acid (HCAF) was introduced. HCAF was covalently linked to the GMA-functionalized CS by reaction in solution using an enzymatic catalyst, laccase, which did not affect the catechol groups responsible for the biological activity of the molecule. The prepared systems were physico-chemically characterized by infrared spectroscopy (FTIR), thermogravimetric analysis (TGA), electron microscopy (FESEM), contact angle, swelling, mechanical analysis, and a determination of breathability. The antioxidant activity of the functionalized systems was evaluated by UV-vis spectroscopy. Finally, the most promising systems were subjected to antimicrobial tests (Kirby Bauer test and dynamic contact test) to evaluate their possible application in wound healing.

## 2. Results and Discussion

The development of systems that can heal a high percentage of wounds quickly, completely, and sustainably is of paramount importance in daily clinical practice. This necessity has led researchers to develop bioactive membranes, i.e., systems capable of encapsulating components that can have a positive action on the healing process, limiting inflammatory phenomena [12]. In the realization of these devices, polymers of natural origin are widely used, thanks to their biocompatibility, absence of toxicity, and biodegradability [46]. Furthermore, they are preferable to synthetic polymers because of the presence of a high number of functionalities, which are of crucial importance in the binding and subsequent release of bioactive molecules. Among the most used polymers of natural origin, CS is the most suitable for making such systems, thanks to its biological properties. In fact, although the mechanism of action has not yet been fully defined, CS has excellent antimicrobial characteristics, probably thanks to the positive charge of its amino groups, which can interact with the negatively charged bacterial barrier [16]. However, as reported by Goy et al., this property is also dependent on the molecular weight of CS [17]. In particular, depending on the gram-(+) or gram-(−) nature of the bacteria, the effect of the molecular weight is different. A further property that allows for CS’ use in TE is its ability to promote the formation of granulation tissue, as it can modulate the migration of neutrophils and macrophages by modifying the subsequent tissue repair processes, thus allowing for the formation of collagen fibers [47]. CS also exhibits antioxidant properties due to its -OH groups [48]. However, these latter features are not sufficient to limit the phenomenon of oxidative stress that occurs during the inflammatory phase [49]. The high production of ROS can, in fact, prolong the inflammatory phase by blocking the healing process. For this reason, the introduction of antioxidant molecules into the dressing can be a valid strategy to improve the antioxidant properties of CS [41,50,51]. As a polysaccharide, CS is generally characterized by poor dimensional stability in an aqueous environment and mechanical properties that are not suitable for the fabrication of systems to be used in TE. For this reason, in the design of CS-based bioactive membranes, the introduction of plasticizing agents and the application of cross-linking processes play an important role. In fact, CS is characterized by an intermolecular and intramolecular network of hydrogen bonds, which limits the mobility of the chains, decreasing the elasticity of the material. The formation of hydrogen bonds between the CS chains and the plasticizer could reduce interactions among the CS chains. For example, as reported by Campos et al., the use of sorbitol as a plasticizing agent enhanced the elongation at break of CS-based dressings from 18% to 83% [33]. However, as reported by Ma et al., the effect of the plasticizer on the properties of dressings was directly dependent on its concentration [52]. In fact, the introduction of the plasticizer can also modify its behavior in physiological systems. As reported by Cai et al., the use of glycerol permitted the increase in the dimensional stability of CS membranes in an aqueous environment [53]. In particular, as the concentration of GLY increased, a notable decrease in the water absorption capacity was observed. Indeed, as GLY is a tri-functional molecule, only the formation of a single bond of GLY with the matrix can cause an effective plasticizing effect (see Figure 1) [54]. 

However, the remarkable hydrophilicity of CS causes cross-linking of the matrix to become necessary. Generally, the cross-linking processes involve the use of the CS amino groups, as in the case of Silva et al., who reported the use of glutaraldehyde as a cross-linking agent, or of Hafezi et al., who instead crosslinked CS using genipin [55,56]. However, since the antimicrobial properties of CS depend on the amino groups, the development of crosslinking reactions that exploit -OH groups is attracting more and more interest. The use of agents such as poly(ethylene glycol) diglycidyl ether (PEGDE) or the introduction of functionalities in the polymer chain that is suitable for the subsequent covalent cross-linking seems to be favored [25]. In particular, the employment of long-chain and flexible cross-linking agents could lead to good mechanical and exudate absorption properties in the dressings. 

Therefore, this study aimed to use solvent casting to fabricate membranes based on CS modified with GMA and GLY and then crosslinked with EGDMA. GMA was used to introduce double bonds to the -OH groups of the polymer (see Figure 2a), which are then exploited in the subsequent cross-linking reaction with EGDMA (Figure 2b) [57] and GLY to provide a good degree of elasticity to polymeric membranes. To evaluate how the crosslinking process and the plasticizer that is used affect the mechanical properties of the films, different concentrations of EGDMA and GLY were studied. As for GLY, a significant improvement in the elasticity of the membranes following the introduction of the plasticizer was highlighted (see Figure 2c). However, for quantities above 20% GLY, the matrix was too fragile, and therefore not suitable for applications in tissue regeneration (Figure 2c III). Therefore, to make membranes bioactive, a subsequent functionalization/imbibition reaction with HCAF was carried out on the system containing 20% plasticizer. In the case of covalent immobilization, an FD of 8% was determined, corresponding to 0.48 mmoles of HCAF; while, for the imbibition process, a loading of 0.75 mmoles was found. All the produced samples, along with their relative acronyms, are reported in Table 1.

### 2.1. Infrared Spectroscopic Analysis

The successful introduction of GMA was verified by ATR-FTIR spectroscopy. The spectra of the developed membranes are shown in Figure 3A. In the CS spectrum, it was possible to note absorptions between 3500 and 3000 cm^−1^, corresponding to the -OH and -NH stretching; in the range 2920–2875 cm^−1^, absorption corresponded to the C-H stretching; at 1650 cm^−1^, a peak was found relative to the C=O stretching (amide I); the absorption band at 1560 cm^−1^ was attributed to the amide II (N-H in-plane deformation coupled with C N stretching) and to the N-H bending of the primary amine. Finally, between 1150 and 1000 cm^−1^, absorptions corresponding to the pyranose ring were visible, while at 895 cm^−1^, absorptions were due to the C-O-C and C-O-H bending. Following the introduction of GMA, the formation of a new peak at 1710 cm^−1^ due to the C=O stretching of the esters [58], and the increase in the peak at 1635 cm^−1^ due to the C=C stretching, were observed. Furthermore, the band at 1170 cm^−1^ was sharper owing to the contribution of GMA, as reported by Zhao et al. [59]. The degree of CS acrylation was determined in our previous work using 1H-NMR and was found to be 20–22% [57]. In the case of GLY, the bands at 1170 cm^−1^ and 1180 cm^−1^ increased as the glycerol concentration increased. The enhancement of the band at 1710 cm^−1^ was caused by the formation of H-bonds in the structure by the plasticizer [60], while that of the band at 1180 cm^−1^ was due to the contribution of the C-O stretching of the tertiary alcohols introduced with the glycerol. 

Figure 3B reported the ATR-FTIR spectra of the crosslinked matrices. It was possible to see a notable decrease in the peaks at 1710 cm^−1^ and 1635 cm^−1^ due to the disappearance of the double bonds used for the formation of the crosslinks among polymer chains, in agreement with that reported by Chiu et al. [61]. In addition, a decrease in the band at 1180 cm^−1^ and an increase at 1025 cm^−1^, both due to the pyranose ring, were observed. Probably, the reaction with EGDMA caused a reduction in the number of H bonds formed by GLY, thus decreasing the elasticity of the dressing. Therefore, the most suitable membrane for the development of bioactive systems for TE was found to be the one cross-linked with the lowest concentration of EGDMA. The spectra of the HCAF-containing bioactive membranes, shown in Figure 4, confirmed the introduction of the antioxidant into the CS_GMA_GLY20_0.05 matrix during both physical and covalent bonding. In particular, an increase in the intensity of the band at 1525 cm^−1^, related to the aromatic C-C stretching introduced with the catecholic structure, and a splitting of the band at 1020 cm^−1^ were observed, attributed to the in-plane and out-of-plane –C–C bending of the aromatic ring [62].

### 2.2. Scanning Electron Microscope Analysis

To study the morphological changes in the systems following the chemical modifications, the membranes were observed by scanning electron microscopy (SEM). From the micrographs shown in Figure 5, it was possible to note that, after the introduction of GMA and GLY, the morphology of the membranes remained unchanged compared to that made with CS alone, unlike the crosslinked ones, which presented a fair surface roughness for EGDMA concentrations of 0.5 mM (see Figure 5D), highlighted by a slight shrinkage of the film. In contrast, the CS_GMA_GLY20_0.05 sample showed a similar surface structure of CS (see Figure 5C). For this reason, this concentration was chosen to produce crosslinked membranes. Furthermore, the introduction of HCAF, both covalent and by imbibition, did not cause morphological changes in the membranes, which showed a smooth surface (see Figure 5E,F). Since images of the membranes at higher magnifications did not show the presence of micro- or nanoporous structures, the systems were subjected to porosity measurement using the liquid displacement method.

### 2.3. Thermal Analysis

Thermal characterization was carried out to obtain information on the CS functionalization reactions. In Figure 6A, the thermogravimetric curves of the non-crosslinked membranes (CS_GMA and CS_GMA_GLYX) are reported, while in Table 1, the degradation temperatures at the maximum mass loss (T_D_) of the membranes can be observed. The GMA introduction left the degradation temperature of CS almost unchanged. Instead, the subsequent introduction of GLY caused a decrease in T_D_, which was very marked in the case of the sample containing 30% (*w*/*w*) of GLY. This phenomenon was due to the plasticizing effect of GLY (interposition of GLY molecules between the chains of CS), which caused a decrease in the ordered regions in the structure of the pristine CS. Furthermore, the introduction of GLY caused an increase in the temperature of the first weight loss, relative to the absorbed water around 110–120 °C, and the formation of a further weight loss between 170 and 200 °C due to the degradation of GLY, as reported by Vázquez et al. [63].

As the matrix containing 30% GLY was too fragile (as seen in Figure 2c), the cross-linking process was only performed on the matrix containing 20% GLY. Figure 6B shows the cross-linked sample thermograms. Generally, thermal behavior is always dependent on the cross-linking process. In this case, it was possible to observe an increase in the degradation temperature with the cross-linking process, which also led to a decrease in the amount of water absorbed by the membranes (a decrease in the value of the first weight loss). This latter phenomenon was more evident for the highest concentration of EGDMA. In this sample, in fact, the first weight loss due to GLY was almost absent (170–200 °C), confirming that as the concentration of the cross-linker increased, a denser structure was formed, which limited the mobility of the CS chains and the formation of H bonds between GLY and CS. From these data, high concentrations of EGDMA were considered unsuitable for obtaining an elastic and flexible membrane. Therefore, the CS_GMA_GLY20_0.05 sample, crosslinked with the lower concentration of EGDMA, was chosen for subsequent functionalization with HCAF.

As far as the thermal stability of the bioactive matrices was concerned, a significant decrease in the degradation temperature was detected (Table 1), as is also observable from the thermograms shown in Figure 7. For both the used antioxidant introduction procedures, it was possible to note that HCAF molecules disturbed the interactions among CS chains by decreasing them, as found in the case of the GLY introduction, leading to a further decrease in T_D_. 

### 2.4. Mechanical Characterization and Evaluation of the Adhesive Properties

As reported by Peh et al. [31], a good dressing must have appropriate flexibility and elasticity to allow for excellent adhesion to the affected area, but also good stress resistance. In fact, the dressing must be able to respond to various stresses depending on the area in which it is used. For this reason, the developed membranes were subjected to tensile measurements. The obtained stress–strain curves are reported in Figure 8A,B, while in Table 1, the values of elastic modulus (E), elongation at break (%), and stress at break can be observed. Pristine CS exhibited unfit behavior for a dressing; the elastic modulus value was too high while the elongation at break was too low. Therefore, chemical and physical modifications were required to develop a suitable dressing. The use of GMA significantly decreased the elastic modulus by increasing the flexibility of the material (elongation at break = 95%). This was due to the lower cohesion among the CS chains caused by the introduction of GMA. This behavior was even more evident after the plasticizer’s introduction. Indeed, the membranes in which GLY was present showed a decrease in the elastic modulus of two orders of magnitude, associated with an enhancement in flexibility with increasing plasticizer concentration. As reported by Ma et al. [52], the formation of the H bonds between the plasticizer and CS chains significantly reduced the brittleness of the membrane made of only CS.

The concentration of GLY that ensured the greatest elongation at break was 20% (*w*/*w*). Therefore, this matrix was the most suitable for the manufacture of dressings. From the stress–strain curves shown in Figure 8B, it was possible to notice how the cross-linking process significantly modified the mechanical properties of the CS_GMA_GLY20 membrane. As the concentration of the cross-linking agent increased, a remarkable increase in the value of the elastic modulus, associated with a decrease in the flexibility of the matrices, was observed (see Table 1). However, the concentration that allowed for good values of elongation at break to be maintained, with associated good values of stress at break, was 0.05 mM. This confirmed that as the concentration of EGDMA increased, the mobility of the CS chains decreased due to the formation of a denser structure limiting the matrix elongation (Table 1). This was in accordance with R. Panchel, who reported that the covalent cross-linking of the CS-PVA dressing with genipin produced a more compact medication with less elongation at rupture [64]. For this reason, the most suitable concentration of EGDMA was considered to be 0.05 mM.

In accordance with what was previously highlighted by thermal analysis, the bioactive membranes showed different behaviors. Specifically, by observing the stress–strain curves shown in Figure 9 and the data reported in Table 1, it was possible to notice a loss of flexibility in the matrices associated with a higher elastic modulus value. In the case of the membrane covalently functionalized with HCAF, the elongation at break was reduced by 30%, confirming the formation of a rather dense structure, due not only to the chemical bond but also to the numerous hydrogen bonds formed between the polymer and antioxidant molecule, which reduced the mobility of CS chains. 

As for the HCAF imbibed matrix, a 10% decrease in the elongation at break was observed. In this case, since GLY was introduced before functionalization with HCAF, it could generate a greater amount of H bonds and consequently exert a stronger plasticizing effect. By comparing our systems with commercial polysaccharide-based dressings, it was possible to confirm the high applicability in the biomedical field of the bioactive membranes developed in this work. In fact, as reported by Minsart et al., dressings based on polysaccharides, such as Aquacel ^®^ Ag Hydrofiber dressing, composed of sodium carboxymethylcellulose (CMC) with silver ions incorporated, or Kaltostat ^®^ dressing, based on alginate fibers which, when in contact with wound exudate, form a firm solid gel/fiber, showed elongation at break values of less than 100%, comparable with those found in the present study [65]. However, in our case, the systems showed a much higher elastic modulus value than commercial dressings, confirming the production of more resistant and long-lasting matrices.

The adhesive performance of the developed bioactive membranes was studied by a lap shear test using fresh pig skins. In Figure 9B, the values of adhesive strength, expressed as a detachment force, of the CS_GMA_GLY20_0.05, CS_GMA_HCAF_GLY20_0.05, and CS_GMA_GLY20_HCAF_0.05 samples were reported. It was possible to notice that the presence of HCAF enhanced the detachment stress value of the bioactive membranes with respect to the free-catechol one, from 30 to 60%. This confirmed that the catechol moieties of the antioxidant could form covalent bonds with functional groups present in skin components. Furthermore, the tissue adhesion properties of the bioactive membranes increased with the increase in HCAF concentration (see the data reported in the results and discussion section). Indeed, the system containing the imbibed antioxidant showed a higher increase in the detachment stress value compared to that with covalently bound HCAF (1.8 kPa vs. 1.4 kPa). These results were in accordance with those obtained in other literature studies [42,43,44,45]. 

### 2.5. Water-Absorption Kinetics 

In dressing development, the ability to absorb water is of paramount importance. A dressing must, in fact, rebalance the absorption and release of liquids, allowing the wound to heal properly. Furthermore, a dressing must not change its morphology too much following the absorption of the exudate [66]. In Figure 10A, the swelling kinetics of the pristine CS and its derivatives with GMA and GLY are reported. CS, being a polysaccharide, was characterized by a high dimensional instability in an aqueous environment, associated with a high swelling capacity. Observing the curves, it was possible to notice that the introduction of GMA tended to significantly reduce the swelling of the matrices compared to the pristine CS. This behavior confirmed the chemical modification of the CS, which also involved the formation of hydrogen bonds between the CS chains. The swelling of the membranes further decreased with the introduction of GLY, particularly with an increase in the concentration of the plasticizer. This was attributable to the formation of a hydrogen bond network, as reported by Cai et al. [53].

Regarding the crosslinked matrices (Figure 10B), as the concentration of EGDMA increased, the swelling capacity of the membranes reduced, reaching a plateau more quickly at higher crosslinker concentrations (0.1 and 0.5 mM). Hence, the kinetics of swelling also confirmed that the cross-linking reaction performed with 0.05 mM EGDMA presumably facilitated the creation of a less compact structure, providing an environment in which GLY could effectively exert its plasticizing effect.

The introduction of HCAF significantly changed the ability of membranes to absorb water. For example, Wang et al. found that the CS-based hydrogel modified with 3,4 dihydroxyhydrocinnamic acid was more stable than the non-functionalized analogue [67]. This was attributed to oxidation phenomena affecting the catechol, which made the CS chains more crosslinked, increasing the hydrophobic character of the matrix. In our case, it was evidenced that these matrices reached the absorption plateau in 20–30 min, unlike the non-functionalized matrix (50 min) (Figure 11). These data confirmed the formation of a more cohesive and dimensionally stable membrane structure in an aqueous environment. 

Furthermore, in accordance with the mechanical properties, the CS_GMA_HCAF_GLY20_0.05 matrix showed the lowest swelling value, while the CS_GMA_GLY20_HCAF_0.05 matrix was characterized by a greater swelling, although this was still less than the non-functionalized matrix. In both cases, the two membranes were stabilized by functionalization with HCAF.

### 2.6. Static Contact-Angle Measurements

As already stated, the ability of a dressing to absorb excess exudate is of fundamental importance in correct wound healing. A membrane with hydrophobic features is not able to soak up exudate, while one that is too hydrophilic absorbs it all. Membrane wettability, determined by contact angle measurements, can provide valid information on the hydrophobic or hydrophilic character of the system. Contact angle values (θ) higher than 90° evidence the hydrophobicity of the matrix, while, with values lower than 90°, the membrane is hydrophilic. Moreover, it must be considered that wettability also plays a fundamental role in defining the biocompatibility of a system. Achieving a good balance between hydrophilic and hydrophobic characteristics can lead to a better interaction of the injured tissue with the dressing, limiting its painful removal [68]. The θ values of prepared membranes are shown in Table 1, while the images of some selected samples, as an example, are reported in Figure 12.

Pristine CS was characterized by a contact angle of 85° ± 0.8°, indicating the hydrophilic character of the polysaccharide due to the presence of OH and NH_2_ groups. The subsequent introduction of GMA and GLY caused a significant decrease in the contact angle value, resulting in a highly hydrophilic membrane, as reported by Ma et al. [52]. However, as expected, the cross-linking process, significantly increased the θ values, especially when high concentrations of EGDMA were used. These results are in agreement with what was reported by S. S. Silva et al. who, after cross-linking CS-based dressings with genipin, found a significant increase in the hydrophobic characteristics of the developed dressings [69]. The exception was the 0.05 mM EGDMA concentration, where the matrix maintained a more hydrophilic character (86° ± 0.6°). Regarding the other two concentrations of EGDMA that were used, the membranes were hydrophobic (θ ≥ 90°), confirming that the cross-linking reaction, limiting the mobility of CS chains, did not allow for the correct implementation of the plasticizer activity and interaction with water molecules, unlike the matrix with the lowest cross-linking agent concentration.

As far as bioactive matrices were concerned, a slight increase in the contact angle was observed in the case of covalent functionalization. Indeed, the HCAF imbibed membrane showed an almost unchanged θ value. In any case, the introduction of the antioxidant led to matrices with a good balance between hydrophilic and hydrophobic character.

### 2.7. Water Vapor Transmission Rate (WVTR) and Porosity Measurements

A further index of the ability of a dressing to interact with the exudate, thus keeping the wound hydrated, is the water vapor transmission rate (WVTR). A dressing is able to retain moisture when its WVTR is <840 g/m^2^/d [70]. The WVTR depends strongly on the porosity of the membrane. In fact, high water vapor permeability in the membrane (large pore size) increases the rate of dehydration at the wound site, while its low porosity does not protect the wound from the excessive adsorption of exudate. Both cases may lead to improper tissue healing. Table 1 shows the WVTR values of all the investigated samples. In agreement with what was seen with the wettability measurements, the introduction of GMA led to an increase in WVTR, evidencing an enlargement of the CS pores due to the presence of GMA in the side chains. In fact, the porosity measurements performed with the liquid displacement method confirmed an increase in the ε value from 50 ± 3 to 65 ± 2% for CS and CS_GMA membranes, respectively. In addition, the subsequent introduction of the plasticizer significantly increased the WVTR, as was also found by Ma et al. [52]. This increase was directly related to the increase in GLY concentration. In fact, the sample containing 30% (*w*/*w*) of GLY was characterized by the highest WVTR value (890 ± 12 g/m^2^/d) and a porosity of about 80 ± 3%. Despite the good porosity value, the WVTR value was too high for a good dressing. Therefore, this membrane was considered to be not beneficial for wound healing.

The cross-linking reaction led to a reduction in WVTR, confirming the production of a denser and consequently less permeable matrix. The reduction was greater when the cross-linking agent concentration was high. In accordance with this, as the concentration of EGDMA increased, a decrease in porosity was also found, showing values of 68 ± 1% for CS_GMA_GLY20_0.05, 45 ± 3% for CS_GMA_GLY20_0.1 and 35 ± 2% for CS_GMA_GLY20_0.5. 

Healthy skin has a WVTR of 204 g/m^2^/d, and following damage processes the limit value rises to 279 g/m^2^/d. However, depending on the wound type, the WVTR can reach even higher values [70]. Therefore, since the crosslinked matrices with the highest EGDMA concentrations had a lower WVTR than the limit value (Table 1) and showed the lowest porosity values, they were not suitable for the manufacture of a dressing, while the one crosslinked using a 0.05 mM EGDMA concentration could be used for wound healing applications. The introduction of the antioxidant made the matrix more cohesive, resulting in a decrease in WVTR and porosity value, particularly in the case of covalently bonded HCAF, in which an ε value of 58 ± 2% was achieved. As for the matrix containing the absorbed antioxidant, the tendency of HCAF to interact with CS chains was confirmed, limiting CS–plasticizer interactions and resulting in a slight decrease in the porosity of up to 62 ± 3%. However, the membranes prepared in the present work reported WVTR values comparable to those of previously marketed dressings. In fact, as reported by Minsart et al., dressings such as Hydrosorb, based on hydrocellular polyurethane–polyurea hydrogel, or Mapilex Ag, made up of polyurethane foam and a layer of silicone in contact with the wound, capable of exerting antimicrobial activity thanks to the presence of silver nanoparticles, have reported WVTR values of around 300–500 (g/m^2^/d) [65]. These values appeared to be compatible with the values reported for the bioactive samples in Table 1. 

### 2.8. Determination of the Antioxidant Activity

The antioxidant activity of CS is attributed to the hydrogen-donating ability of hydroxyl or amino groups of the polymer. The active hydrogens of the polysaccharide can react with ROS to form stable macromolecular radicals. The scavenging properties of CS also depend on molecular weight and the ability of amino groups to chelate metals [71]. Furthermore, the CS antioxidant activity can be enhanced by its derivatization, as reported in the literature [72]. For example, the functionalization of CS with ferulic acid using EDC:NHS reagents, as reported by Balasubramaniam et al., allowed for values around 90% to be reached, and for capture of the DPPH free radical, confirming the effectiveness of such modifications [73]. Also, Wang et al. reported that, after the conjugation of CS with different amounts of 3,4 dihydroxyhydrocinnamic acid, again using EDC:NHS, hydrogels capable of blocking up to 80% of the free radical DPPH, could be obtained [67]. In this study, the EC_50_ value was used to evaluate the antioxidant properties of the developed systems as it allows for different systems to be compared each other, as reported by Shahidi and Zhong [74]. EC_50_ values of 95 g/L and 0.02 g/L were found for the pristine CS membrane and HCAF, respectively. The cross-linking reaction and the plasticizer introduction reduced the scavenging activity of the CS_GMA_GLY20 matrix by increasing the EC_50_ value by one order of magnitude (500 g/L). By observing the curves shown in Figure 13, it was possible to verify that both methods of introducing HCAF were effective in making the membranes more bioactive. In particular, in the case of the matrix imbibed with HCAF, the EC_50_ was reduced by two orders of magnitude (4 g/L) as the antioxidant was free to interact with DPPH radicals, while this was reduced by only one order of magnitude in the case of the matrix containing the covalently bound antioxidant (12 g/L). 

### 2.9. Antimicrobial Activity

The growing problem of the antibiotic resistance of some pathogens and the appearance of micro-organisms resistant to antibiotics has promoted studies on the development of antimicrobial polysaccharide systems containing natural products, such as antimicrobial peptides or antioxidants. For example, injectable polysaccharide hydrogels for the controlled release of incorporated nisin were investigated by Flynn et al. [75]. Such hydrogels, composed of oxidized dextran, alginate functionalised with hydrazine groups, and glycol chitosan in different percentages, exhibited antimicrobial activity against *S. aureus* for up to 10 days. In addition, it was found that glycol chitosan exerted a synergistic action with nisin in the inhibition of bacterial growth. Lee et al., instead, developed chitosan-based systems conjugated with hydroxycinnamic acids, including caffeic acid, ferulic acid, and sinapic acid [76]. The conjugates showed an improved antimicrobial activity, in terms of minimum inhibitory concentration (MIC), against two standard methicillin-resistant Staphylococcus aureus (MRSA) and foodborne pathogens compared to that of unmodified chitosan. In a very recent study, it was demonstrated that tannic acid (TA), contained in a multifunctional scaffold composed of Schiff-base-crosslinked konjac glucomannan/chitosan hydrogel, can stabilize the system structure, modulate its degradation, and act as an active drug exerting antioxidant, antibacterial, and anti-biofilm effects [77]. 

To evaluate the effectiveness of our systems in counteracting infections, and therefore promoting tissue regeneration, the bioactive membranes were subjected to preliminary assays against *E. coli* (gram-negative) and *S. aureus* (gram-positive) using the agar diffusion method, and a test to determine the reduction in bacterial growth (method E2149-20). There were no zones of inhibition against these bacteria, as seen in Figure 14, not even for films with imbibed HCAF that possesses a greater amount of the active compound. However, a migration of the active component from the films to the agar plate was clear, even in the covalently attached HCAF (see the brown HCAF and around the films). 

Therefore, since this effect was clearly observed, a second test using the dynamic contact between the strains and films was used to evaluate the antimicrobial activity of the films. The strain alone was used as blank control and CS and CS_GMA_GLY20 were used as a negative control. When the initial bacteria concentration in contact with the films was ~10^5^ CFU/mL, both the HCAF-functionalized films were able to kill the bacteria with 99.999% effectiveness, while CS did not have any effect against bacteria and CS_GMA_GLY20 provoked an increase in the population in one order. In order to evidence the differences between films with covalently or physically attached HCAF, the same assay in broth solutions was performed by increasing the bacterial concentration by up to ~10^8^ CFU/mL. The results showed that while the CS_GMA_HCAF_GLY20_0.05 film was not able to kill *S. aureus*, the CS_GMA_GLY20_HCAF_0.05 film was capable to destroy the strain population with a rate of around 50%. In the tests against *E. coli* bacteria, the response was about 40% in both cases.

## 3. Materials and Methods

### 3.1. Materials

CS from shrimp shells with medium molecular weight, 200–800 cP and 75–85% deacetylation degree, and glycidyl methacrylate (GMA) at 97%, containing 100 ppm monomethyl ether hydroquinone as an inhibitor, GLY at 99.5%, and ethylene glycol dimethacrylate (EGDMA) at 98%, which contains 90–110 ppm monomethyl ether hydroquinone as an inhibitor and 3,4 dihydroxyhydrocinnamic acid (HCAF) at 98%, 2,2-diphenyl-1-picrylhydrazyl (DDPH), and a dialysis tubing cellulose membrane (cut off: 14,000 Da) were purchased from Sigma Aldrich. For antimicrobial tests, sodium chloride solution (NaCl suitable for cell culture, BioXtra, Welwyn, UK) and phosphate-buffered saline powder (PBS, pH 7.4) were used, both from Sigma-Aldrich (San Luis, MO, USA). Cellulose disks (grade AA disk) with 6 mm diameter were purchased by Whatman. Mueller–Hinton and Columbia (5% of sheep blood) agar plates were purchased from Deltalab (Barcelona, Spain). American Type Culture Collection (ATCC, Manassas, VA, USA) *Staphylococcus aureus* (*S. aureus*, ATCC 29213) and *Escherichia coli* (*E. coli*, ATCC 25922) were acquired from Oxoid (Hampshire, UK).

### 3.2. Preparation of CS_GMA and CS_GMA_GLYX Membranes

To obtain membranes based on chitosan GMA-functionalized CS, the procedure reported by Silvestro et al. was used, with some modifications [57]. In brief, 0.5 g of CS was dissolved in 50 mL of 2% *v*/*v* acetic acid to obtain a 2% (*w*/*v*) solution. After 24 h of stirring, the solution was poured into a flask and GMA was added in a ratio of 2:1 to the repeating unit of CS. GMA was introduced dropwise, together with a small amount of hydroquinone as a stabilizer. The reaction was carried out in an inert atmosphere (N_2_) at the temperature of 70 °C for 24 h, with constant stirring at 300 rpm. Afterward, the polymeric solution was dialyzed for 2 days, with continuous water changes, to remove any unreacted molecules. Once the dialysis was completed, GLY, selected as a plasticizer, was introduced using weight ratios of 10%, 20%, and 30% (*w*/*w*) with respect to CS. The solutions were then left under stirring for 4 h to homogenize and allow for the formation of hydrogen bonds. Subsequently, they were poured into Petri dishes (d = 8 cm) and left at room temperature for 2 days to allow for solvent removal and to obtain dry membranes. The acronyms used for the systems were: CS for pristine chitosan, CS_GMA for modified chitosan, and CS_GMA_GLYX for the system containing glycerol, where X was the percentage of GLY used for the formation of the three different dressings.

### 3.3. Chemical Crosslinking of CS_GMA_GLYX Matrices

To obtain dimensional stability among the systems in an aqueous environment and to investigate how the cross-linking process influenced the properties of the membranes, the cross-linking reaction was carried out in the solid phase, by dipping the membranes in water solutions of EGDMA monomer at different concentrations (0.05, 0.1, and 0.5 mM). Subsequently, sodium metabisulfite (SMBS) and potassium persulfate (PPS), in a 1:1 ratio and a ratio of 1:10, with respect to the moles of EGDMA as initiators of the reaction, were introduced. The reaction was carried out at room temperature for 10 min. Then, the obtained membranes were repeatedly washed with distilled water to remove the unreacted molecules. Finally, the matrices were dried at room temperature for 2 days. The acronyms used for the systems were CS_GMA_GLYX_Y, where X was 20% (*w*/*w*) of GLY and Y referred to the monomer concentration of EGDMA used with respect to the CS containing the GMA monomer.

### 3.4. Introduction of 3,4-Dihydrocinnamic Acid

The introduction of HCAF into the CS-based systems was accomplished through two methodologies. In the first case, slightly modifying the procedure reported by Brzonova et al., the formation of a covalent bond between CS and HCAF was obtained using an enzyme as a catalyst [78]. Once the CS functionalization with GMA was performed, the dialyzed solution was left to react with HCAF. In brief, an amount in grams corresponding to a concentration of 10 mM of HCAF was added to the dialyzed CS_GMA solution (V = 50 mL), together with 0.5 mL of laccase (20 nkat mL^−1^ in 2 mL of 50 mM sodium citrate buffer at pH 4.5). The reaction was left to proceed for 11 h in the dark, maintaining the temperature of 30 °C and stirring at 150 rpm. The kinetics of the reaction was followed by the Ninhydrin test [79], which also allowed for the evaluation of the degree of functionalization (*FD* %). *FD* was determined by the difference of the initial concentration of the amino groups *(C*_2_) and the concentration of those remaining after the reaction with HCAF (*C*_1_) divided by their initial concentration (*C*_2_), as shown in the following equation:$$\%\;FD=\frac{\left(C_{2}-C_{1}\right)}{C_{2}}\times 100$$

At the end of the reaction, the solution was again dialyzed for 2 days to remove unreacted antioxidant. Finally, 20% (*w*/*v*) GLY was added as a plasticizer. In this case, the solution was left under stirring for 4 h and subsequently poured into petri dishes (d = 8 cm). The dry membrane was obtained after evaporation of the solvent at room temperature for 2 days. Then, the matrix was cross-linked with EGDMA by using the previously reported procedure.

The second procedure involved the imbibition of the solid membrane with an HCAF solution. In this case, ionic and hydrogen bond interactions between the antioxidant molecule and CS matrix were exploited. Specifically, the dried CS_GMA_GLYX membrane was immersed in the dark in a 10 mM aqueous solution of antioxidant (V = 150 mL) at room temperature for 2 h. After this time, the matrix was no longer able to absorb the antioxidant. HCAF loading was determined by UV-VIS spectroscopy using a calibration curve obtained at 280 nm. Subsequently, the matrix was washed 3 times with distilled water to eliminate the excess reagent and then underwent the cross-linking process as previously reported. 

The acronyms used for the systems were as follows: CS_GMA_HCAF_GLYX_Y for the membrane with the covalently bound HCAF and CS_GMA_GLYX_HCAF_Y for the one in which HCAF was introduced ionically. In both cases, X was 20% (*w*/*w*) of GLY and Y was the 0.05 mM concentration of EGDMA used for cross-linking the matrix. 

### 3.5. Infrared Spectroscopic Analysis

The chemical and ionic modification of CS was evaluated by infrared spectroscopy (FTIR). All the systems were analyzed in attenuated total reflection (ATR) mode using a Nicolet 6700 (Thermo Fisher Scientific, Waltham, MA, USA) equipped with a Golden Gate single reflection diamond ATR accessory. Specifically, 200 scans and a resolution of 2 cm^−1^ were used to record the spectra.

### 3.6. Scanning Electron Microscope Analysis

To investigate the influence of the CS chemical and ionic modifications on the system properties, the membrane morphology was analyzed by scanning electron microscopy (SEM) using a Philips XL30 microscope (Phillips, Eindhoven, The Netherlands). Specifically, the films were observed both on the surface and in bulk after their fracture in liquid nitrogen. The samples before measurement were sprayed with an 80:20 Au/Pd alloy to allow for observation.

### 3.7. Thermal Stability Analysis

To evaluate the thermal stability of the modified polymer systems, the CS-based membranes were subjected to thermogravimetric analysis (TGA). Measurements were performed in N_2_ atmosphere using 5–7 mg of sample, a temperature range from 30 to 500 °C, and a heating rate of 10 °C/min. The instrument used was a Mettler TG 50 thermobalance (Mettler Toledo, Columbus, OH, USA).

### 3.8. Mechanical Characterization and Evaluation of the Adhesive Properties

The mechanical properties of the systems were evaluated by using an ISTRON 4502 instrument (Instron Inc., Norwood, MA, USA). Specifically, the samples were cut into a rectangular shape (500 × 100 × 0.3 mm, representing length, width, and thickness, respectively). Subsequently, they were placed between the two Instron clamps and subjected to a maximum load of 2 kN at a constant strain rate of 10 mm/min. To determine the elastic modulus, tensile strength, and elongation at break, 5 measurements were carried out for each sample.

To determine the lap shear adhesive strength of the developed systems, the same ISTRON 4502 instrument was used. Two fresh defatted pig skins, employed as a model tissue, were cut into rectangular shape (2.5 cm × 4 cm), inside which the sample membrane (25 × 25 × 0.1 mm, representing, length, width, and thickness, respectively) was positioned in the overlapping area. Then, the lap joint was slightly compressed with paperclips and the device was placed in phosphate buffer solution (PBS, pH = 7.4) at 37 °C for 30 min to simulate the wet environment of the human body. After this time, the two final parts of the pig skin were fixed to the clamps of tensile machine. The measurements, performed using a shear rate of 0.5 mm/min and a maximum load of 500 N, were conducted until the two pig skins separated. The lap shear strength was determined from the maximum of the experimental stress–strain curve and expressed as a detachment force. Each sample was measured five times and expressed as the mean and standard deviation.

### 3.9. Water-Absorption Kinetics

The membranes were subjected to the water absorption measurements necessary for their application. In particular, the films were weighed (*W*_0_) and immersed in a phosphate buffer solution at pH 7.4. At set times, the samples were taken, lightly blotted with filter paper, and weighed (*W_t_*). The water absorption was calculated using the following relationship:$$W\;\%=\left(\frac{W_{t}-W_{0}}{W_{0}}\right)\times 100$$

To determine the standard deviation value, 5 experiments were carried out for each sample.

### 3.10. Static Contact-Angle Measurements

The drop method was used to evaluate the static contact angle of the developed systems. In order to obtain a flat surface, the membranes were initially subjected to constant pressure for 1 day. Then, a picture was taken after 10 µL of Milli-Q water was placed on the surface of the films. The Motic Images Plus 2.0 program was used to process the acquired images, which made it possible to evaluate the length of the fall base (*D*) and the fall height (*h*). The contact angle was determined using the following relationship:$$\theta =2\;\tan^{-1}\frac{2h}{D}$$

Each contact angle value was obtained as the mean value of 5 measurements collected on different points of the surface.

### 3.11. Water Vapor Transmission Rate (WVTR) and Porosity Measurements

To assess the permeability of a material, water vapor transmission rate (*WVTR*) measurements were generally performed. *WVTR* describes the amount of water vapor that passes through a solid material over a set period of time. The *WVTR* was determined by fixing the film onto the opening of a cylindrical flask (diameter 1.8 cm) that was previously filled with 10 mL of distilled water. The flask was left at 37 °C for 2 days (ASTM E-96 [80]), during which time it was weighed at constant intervals. The *WVTR* values were calculated using the following equation:$$WVTR=\frac{W}{A\times t}$$
where *W* is the flask weight (g) at time *t*, *A* is the exposure area (m^2^) of the sample, and *t* is the exposure time (d). The mean values and the standard deviation of 5 samples for each film were reported.

As regards porosity measurements, a liquid displacement method was used. To avoid swelling of the samples, ethanol was selected as the liquid. A sample with weight *W*_0_ and volume *V*_0_ was immersed in 5 mL of ethanol (density 0.789 g/cm^3^ at 20 °C) for 30 min. After that time, the membrane was removed and weighted (*W*_1_). The porosity (*ε*%) was determined by employing the following equation: $$ℇ\;\%=\left(\frac{W_{1}-W_{0}}{\rho_{EtOH}\;V_{0}}\right)\times 100$$

Three porosity experiments were performed for each sample and the data were reported as average value ± standard deviation.

### 3.12. Determination of Antioxidant Activity 

The HCAF’s introduction into the membranes was evaluated by measuring the free radical scavenging activity of some selected samples, specifically CS_GMA_HCAF_GLY20_0.05 and CS_GMA_GLY_HCAF_0.05. The procedure involved the use of DPPH as a radical model, according to the method developed by Marsden Blois and subsequently perfected by Brand-Williams [81].

The free radical scavenging activity of the antioxidant alone was measured by preparing HCAF solutions at different concentrations in contact with 4 mL of DPPH at a concentration of 0.2 mM. The solutions were left under stirring for the time necessary to reach the plateau (30 min). Then, their absorbance was measured at a wavelength of 520 nm using a calibration curve obtained by measuring the absorbance of DPPH solutions versus the concentration of the model radical. In addition, the percentage of residual DPPH at the steady state for each antioxidant concentration was also plotted from the calibration curve. The antiradical activity of the HCAF solutions was determined by reporting the percentage of residual DPPH at the steady state as a function of the molar ratio between the antioxidant and DPPH. Specifically, the efficient concentration = EC_50_ (g/L) was defined as the number of antioxidants required to reduce the initial DPPH concentration by 50%. The antiradical activity was then obtained graphically from the value that the curve assumes at 50% DPPH in a steady state.

Differently, the antioxidant activity measurements of the functionalized samples were performed on the solutions containing a small amount of sample. Different quantities of the film (0.2–5 mg) were immersed in 4 mL of DPPH and left under stirring for 30 min. Then, the solution absorbance at 520 nm was evaluated. The EC_50_ was calculated graphically by plotting the percentage of residual DPPH at the steady state versus the amount of sample (g) per volume (L) of DPPH used. Therefore, EC_50_ is equivalent to the grams of sample present in the DPPH solution, expressed in L, which is necessary to obtain 50% residual DPPH at the steady state.

### 3.13. Antimicrobial Activity

The antibacterial activity of the CS-modified membranes was evaluated against Staphylococcus aureus and Escherichia coli, which are both opportunistic pathogens that interfere in the wound healing process. In the first approach, the antimicrobial activity of CS-based films was obtained following the Kirby–Bauer disk diffusion susceptibility test protocol [82], using CS and CS_GMA_GLY20 as blank controls. The strains were grown in Mueller–Hilton agar plates (Fisher Scientific) for 24 h at 37 °C. Then, a concentration of 0.5 McFarland (~10^8^ colony-forming units (CFU)/mL) was obtained using a sterile saline solution (Scharlab). Subsequently, a sterile swab was dipped into the tube containing the bacteria that were to be inoculated on the surface of the agar plate by streaking the swab. Afterward, CS and HCAF were dissolved in water at a concentration of 20 µg/mL; then, 25 µL of each solution was poured into cellulose disks (d = 6 mm) and left to dry. Finally, the cellulose discs were placed on the surface of the inoculated agar plate. Similarly, the films containing the antioxidant were cut into 6 mm disks and placed on the surface of the agar plate. After the incubation of the plates at 37 °C for 24 h, the zone of inhibition around each cellulose disk was measured. 

In addition to the Kirby–Bauer test, the antibacterial activity of the films was measured following the second methodology based on the dynamic contact of the strain broth solution with the films [83]. Fresh bacterial suspensions were prepared with 0.5 McFarland turbidity scale and further diluted to ~10^6^ CFU/mL with PBS. Next, 0.1 mL of the tested inoculum and 0.9 mL of PBS were placed in sterile Falcon tubes, which contained the films with the same dimensions as the previous protocol, to reach a working solution of ~10^5^ CFU/mL. The suspensions were stirred at 120 rpm for 24 h. Subsequently, this solution was diluted by 10-fold serial dilutions in sterile Falcon-style tubes and each dilution was poured onto Columbia 5% Sheep Blood agar plates (Fisher Scientific) and incubated for 24 h at 37 °C. The bacteria were counted and the percentage of killed bacteria was measured by the difference between the CFU of the control sample and the corresponding CFUs of the films. Finally, the same test was repeated using a bacteria working solution of ~10^8^ CFU/mL to better evidence the effective difference between the bioactive dressings.

### 3.14. Statistics

Analysis of variance comparisons were performed using a one-way ANOVA test. Differences were considered significant for * *p* < 0.05. Data are reported as means ± SD.

## 4. Conclusions

In this work, bioactive membranes were obtained by both the chemical and physical modification of CS, with GMA and GLY, respectively. It was evidenced that after functionalization with GMA, the swelling properties of CS decreased while the mechanical ones improved in terms of elasticity thanks to the subsequent introduction of GLY, used as a plasticizer. It was verified that the most suitable GLY concentration to obtain membranes that are potentially applicable as dressings for wound healing with good elongation at break, WVTR, and wettability values was 20% (*w*/*w*). After the cross-linking reaction with EGDMA, carried out at different crosslinker concentrations, an increase in the elastic modulus and a decrease in the elasticity of the matrices were observed. However, it was only possible to obtain matrices with contact angle and WVTR values which satisfied the fundamental requirements for obtaining wound dressings when an EGDMA concentration of 0.05 mM was used. 

To limit oxidative stress phenomena that can slow down the wound-healing process, the antioxidant HCAF was introduced into the membranes by imbibition and by using laccase as a catalyst. Thermal and mechanical analyses confirmed the formation of a more cohesive network, which limited the plasticizing effect of GLY, particularly when HCAF was chemically bound. Despite both membranes maintaining good elongation values at break and good water absorption, the imbibed one showed a lower EC_50_ value, indicating the greater possibility of HCAF interacting with DPPH radicals.

Finally, antimicrobial tests in broth showed the effectiveness of the membrane containing the physically bound antioxidant (CS_GMA_GLY20_HCAF_0.05). These findings seem to be very promising for tissue engineering applications. However, further in vitro and in vivo biological characterizations will be required to better define the potential use of this bioactive system for wound healing.

## Figures and Tables

**Figure 1 ijms-25-01961-f001:**
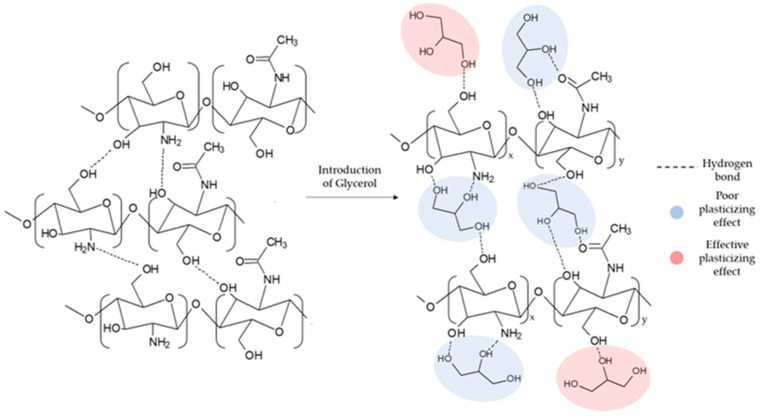
Formation of the hydrogen bonding network responsible for the plasticizing effect of CS.

**Figure 2 ijms-25-01961-f002:**
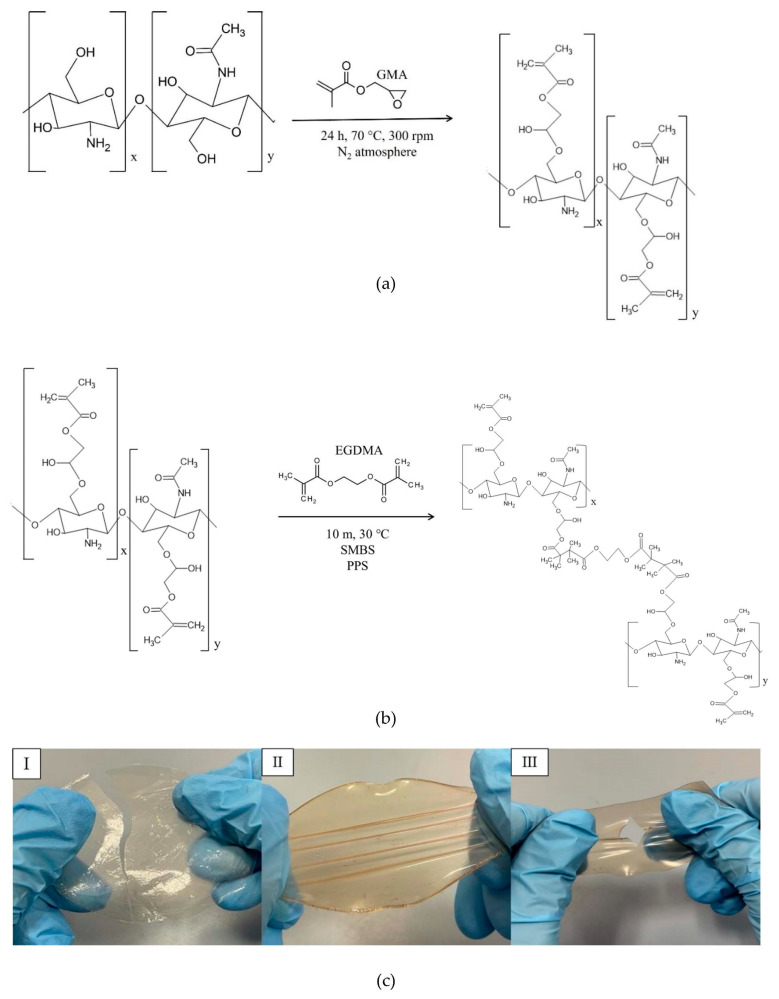
Functionalization of CS with GMA (**a**); cross-linking reaction of CS_GMA with EGDMA (**b**); membranes under stretching (**c**); pure CS (**I**), CS_GMA_GLY20 (**II**) and CS_GMA_GLY30 (**III**).

**Figure 3 ijms-25-01961-f003:**
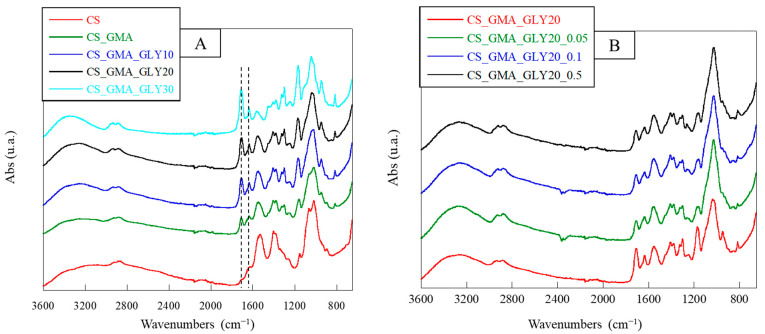
ATR spectra of pristine chitosan (CS), chitosan modified with GMA (CS_GMA) and then with GLY at three different concentrations (10, 20, and 30% (*w*/*w*)) (**A**); ATR spectra of the CS_GMA_GLY20 dressing, and its derivatives, crosslinked using three different concentrations of EGDMA (0.05, 0.1 and 0.5 mM) (**B**).

**Figure 4 ijms-25-01961-f004:**
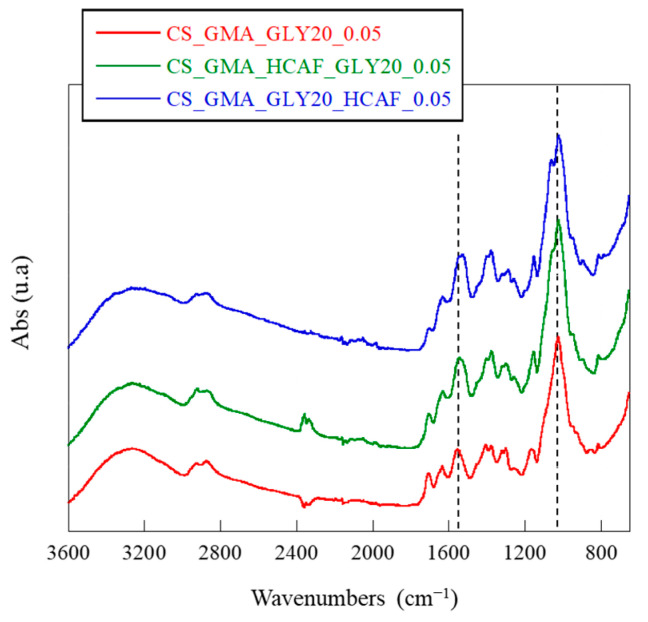
ATR-FTIR spectra of the CS_GMA_GLY20_0.05 membrane and of the same one after being made bioactive after the covalent or physical introduction of HCAF.

**Figure 5 ijms-25-01961-f005:**
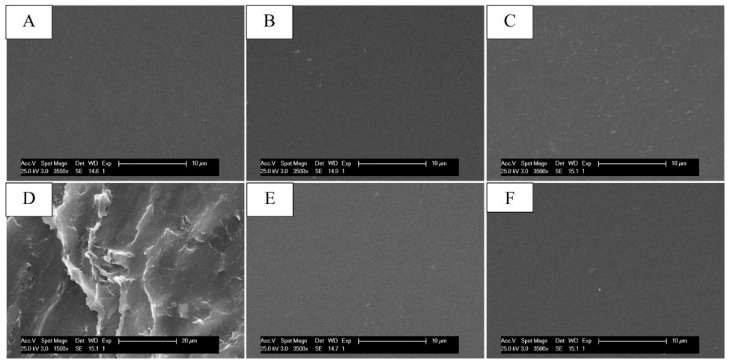
Scanning electron microscopy (SEM) micrographs of the dressings: CS (**A**); CS_GMA_GLY20 (**B**); CS_GMA_GLY20_0.05 (**C**); CS_GMA_GLY20_0.5 (**D**); CS_GMA_HCAF_GLY20_0.05 (**E**) and CS_GMA_GLY20_HCAF_0.05 (**F**).

**Figure 6 ijms-25-01961-f006:**
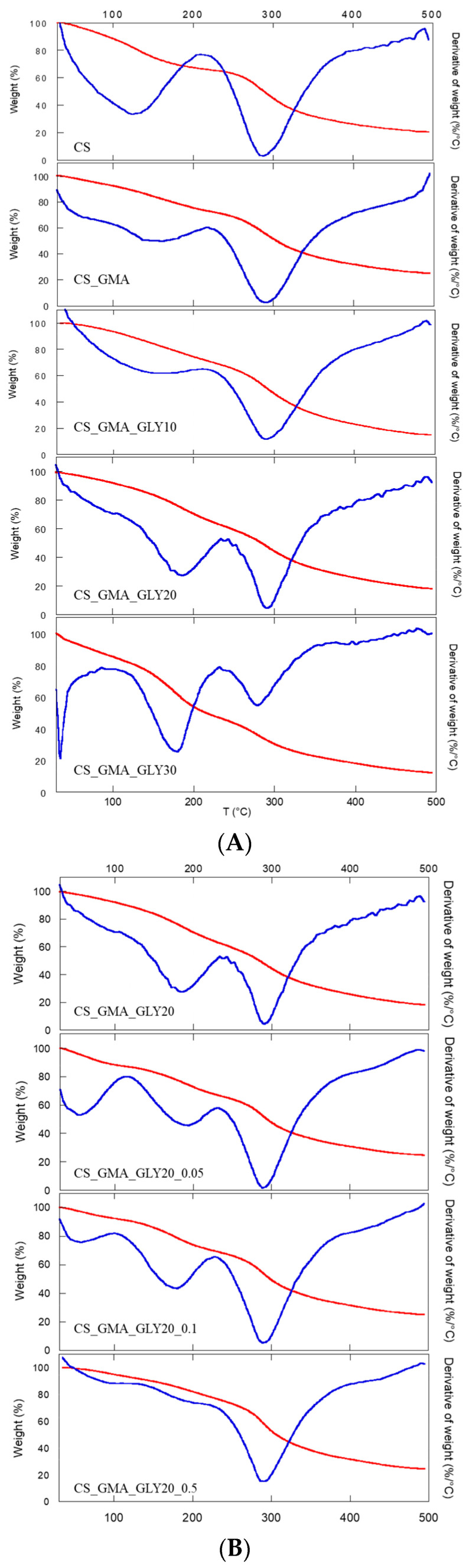
TGA (red line) and DTGA (blue line) curves of pristine chitosan (CS), chitosan modified with GMA (CS_GMA) and with GLY (CS_GMA_GLYX) at three different concentrations (10, 20, and 30% (*w*/*w*)) (**A**); TGA (red line) and DTGA (blue line) curves of CS_GMA_GLY20 dressing, cross-linked using three different concentrations of EGDMA (0.05, 0.1 and 0.5 mM) (**B**).

**Figure 7 ijms-25-01961-f007:**
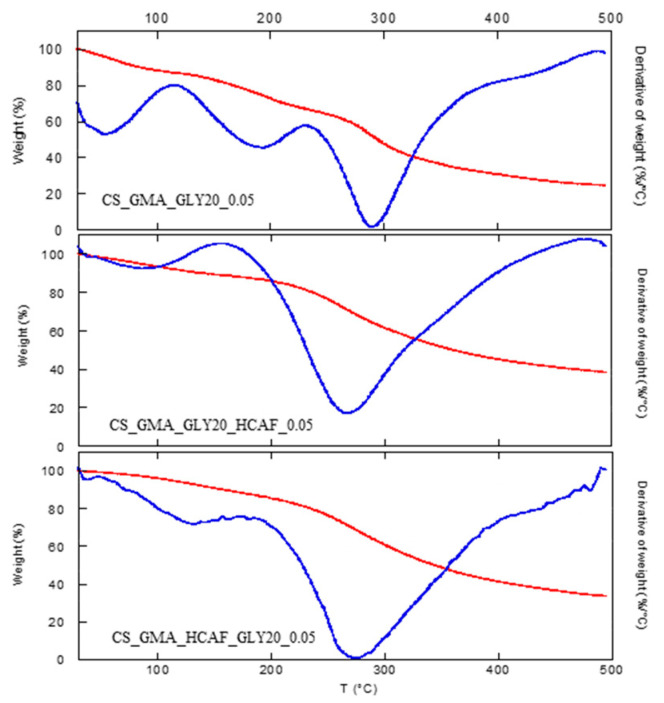
TGA (red line) and DTGA (blue line) curves of the CS_GMA_GLY20_0.05 membrane and the same one after being made bioactive after the covalent or physical introduction of HCAF.

**Figure 8 ijms-25-01961-f008:**
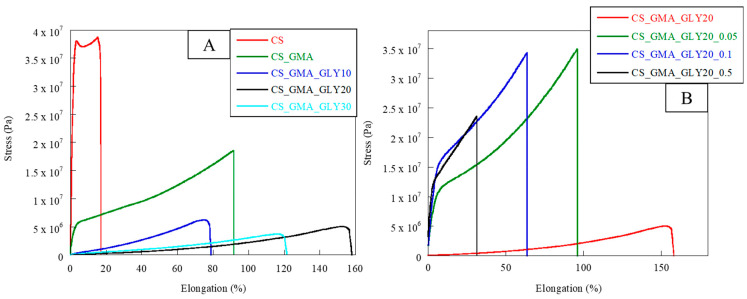
Stress–strain curves of pristine chitosan (CS), chitosan modified with GMA (CS_GMA) and subsequently with GLY at three different concentrations (10, 20, and 30% (*w*/*w*)) (**A**); stress–strain curves of CS_GMA_GLY20 membrane and its derivatives, crosslinked using three different concentrations of EGDMA (0.05, 0.1 and 0.5 mM) (**B**).

**Figure 9 ijms-25-01961-f009:**
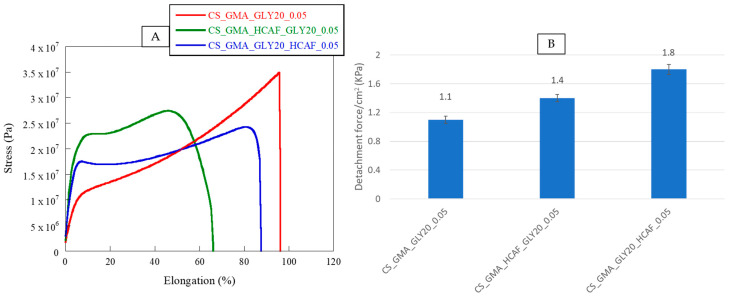
Stress–strain curves (**A**) and detachment force (**B**) of CS_GMA_GLY20_0.05 and bioactive membranes containing covalently or physically bound HCAF.

**Figure 10 ijms-25-01961-f010:**
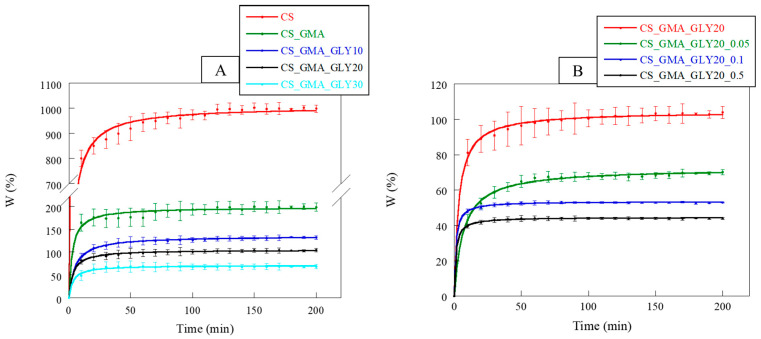
Water-absorption kinetics of the pristine chitosan (CS), chitosan modified with GMA (CS_GMA) and then GLY at three different concentrations (10, 20, and 30% (*w*/*w*)) (**A**); water-absorption kinetic curves of the CS_GMA_GLY20 membrane and its derivatives, crosslinked using three different concentrations of EGDMA (0.05, 0.1 and 0.5 mM) (**B**).

**Figure 11 ijms-25-01961-f011:**
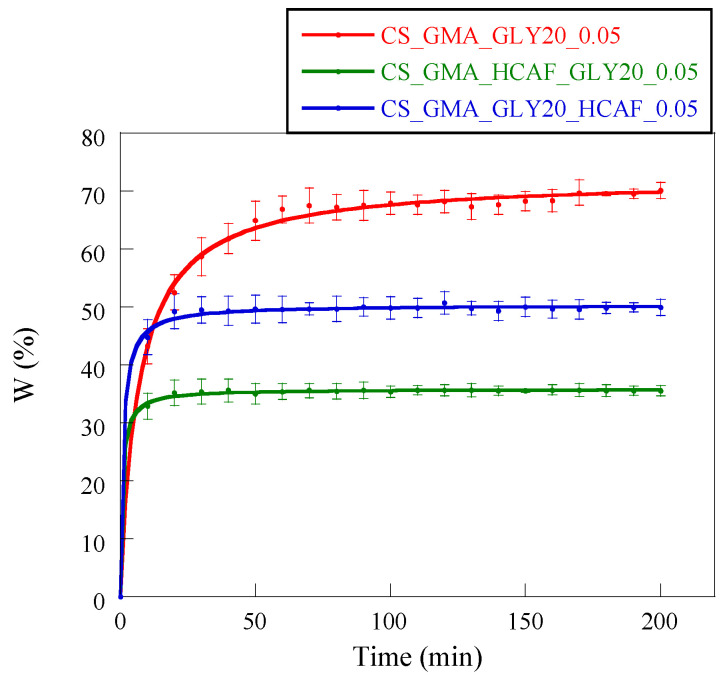
Water-absorption kinetics of CS_GMA_GLY20_0.05 dressing and the bioactive ones obtained after HCAF introduction.

**Figure 12 ijms-25-01961-f012:**
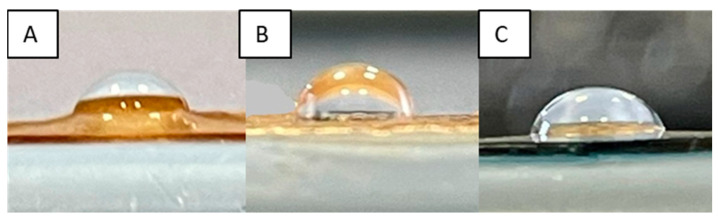
Images of the CS_GMA_GLY20 (**A**), CS_GMA_GLY20_0.5 (**B**), and CS_GMA_GLY20_HCAF_0.05 samples (**C**).

**Figure 13 ijms-25-01961-f013:**
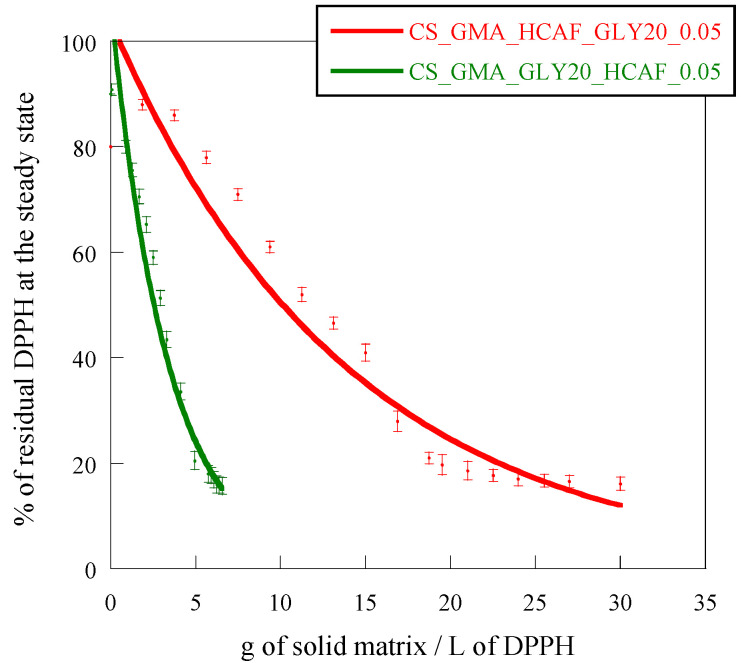
Disappearance of DPPH as a function of g of solid matrix/L of DPPH.

**Figure 14 ijms-25-01961-f014:**
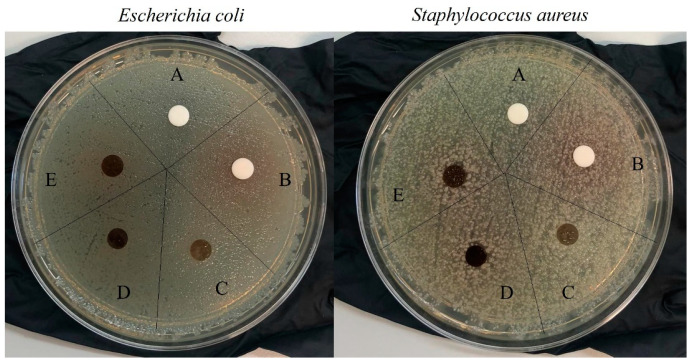
Inhibition halos around CS (A), HCAF (B), CS_GMA_GLY20_0.05 (C), CS_GMA_GLY20_HCAF_0.05 (D), and CS_GMA_HCAF_GLY20_0.05 (E) films.

**Table 1 ijms-25-01961-t001:** Acronyms and physical properties of CS-based membranes. TD = degradation temperature at the maximum mass loss; E = elastic modulus; WVTR = water vapor transmission rate; Θ° = static contact angle. Statistical analysis showed a significant difference when * *p*-value < 0.05.

Acronym	T_D_ (°C)	E (MPa)	Elongation at Break (%)	Stress at Break (MPa)	WVTR (g/m^2^/d)	Θ (°)
CS	290	1890 ± 155	15 ± 3	39 ± 1	598 ± 31	85.5 ± 0.8
CS_GMA	291	184 ± 17	95 ± 4	19 ± 1	620 ± 24	70 ± 1
CS_GMA_GLY10	287	9.2 ± 0.1	77 ± 5	6.5 ± 0.3	647 ± 22	70 ± 1
CS_GMA_GLY20	282	1.7 ± 0.8	148 ± 4	3.2 ± 0.8	760 ± 15	68 ± 2
CS_GMA_GLY30	277	1.9 ± 0.1	130 ± 4	3.5 ± 0.5	890 ± 12	65.4 ± 0.8
CS_GMA_GLY20_0.05	290	280 ± 10	96 ± 8	36 ± 4	450 ± 10 *	86.5 ± 0.6 *
CS_GMA_GLY20_0.1	294	340 ± 16	65 ± 7	33 ± 6	230 ± 26	94 ± 2
CS_GMA_GLY20_0.5	297	480 ± 20	35 ± 2	26 ± 7	185 ± 33	98 ± 1
CS_GMA_HCAF_GLY20_0.05	275	670 ± 30	66 ± 5	27 ± 9	377 ± 17	89.5 ± 0.5 *
CS_GMA_GLY20_HCAF_0.05	265	431 ± 70	87 ± 6	24 ± 8	420 ± 12 *	87 ± 1

## Data Availability

Data will be made available on request.

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
