# Peer review of "Development of Antioxidant and Antimicrobial Membranes Based on Functionalized and Crosslinked Chitosan for Tissue Regeneration"

_ijms, 2024, doi:10.3390/ijms25041961_

Round 1

Reviewer 1 Report

Comments and Suggestions for Authors

1.       There are many anti-inflammatory molecules, why choose 3,4-dihydrocinnamic acid; it is better to explain it in the introduction;

2.       As wound dressing, the bioadhesive strength of e HCAF-imbibed membrane is better to investigate

3.       The title “application in wound healing”, but in the manuscript I do not see any data about wound healing.

4.       The Figure should be improved, for example, Figure 8 the font sizes in the pictures are all different, etc.

5.       In the antioxidant assay, I wounder DPPH can represent free radicals existed in the infected tissue, such as hydrogen peroxide and superoxide?

6.       Figure 13 was not indexed in the manuscript, and from Figure 13, how do you see a 50% reduction in bacteria

7.       In the introduction, some references are better to cited and discussed in the manuscript, for example, European Polymer Journal, 2024, 202, 112592; Biomedical Engineering Communications 2023;2(1):2; Biomaterials Research 27(1):73, etc.

Comments on the Quality of English Language

 Minor editing of English language required

Author Response

Responses to the Reviewer 1.

We thank the Reviewer for agreeing to evaluate the manuscript. We have given significant consideration to the qualified contributions that were found to be fundamental for improving our work's standards of quality and excellence.

  1. There are many anti-inflammatory molecules, why choose 3,4-dihydrocinnamic acid; it is better to explain it in the introduction.

The choice of 3,4-dihydroxicinnamic acid as an anti-inflammatory molecule was due to its ability to increase the adhesion properties of dressings as reported in the sentence on lines 129-132: "As reported by J. H. Ryu et al., catechol-functionalized CS and Pluronic composite hydrogels showed strong adhesiveness to soft tissues and mucous layers, and good haemostatic properties compared to the same catechol-free composite hydrogels." However, to make our choice clearer, the sentence has been changed to:

“As reported by J. H. Ryu et al., 3,4 hydroxycinnamic acid-functionalized CS and Pluronic composite hydrogels showed strong adhesiveness to soft tissues and mucous layers, and good haemostatic properties compared to the same catechol-free composite hydrogels.”

Also in the lines 146-150, the choice for using HCAF has been made clearer.

  1. As wound dressing, the bioadhesive strength of the HCAF-imbibed membrane is better to investigate.

We agree with the Reviewer. Bioadhesive measurements are important to verify the full applicability of the developed membranes in the wound healing field. We used the lap shear test to determine the adhesiveness of our membranes to pig skin employed as a model tissue. In the manuscript have been reported the used experimental procedure (subsection 3.8):

“To determine the lap shear adhesive strength of the developed systems the same IS-TRON 4502 instrument was used. In particular, two fresh defatted pig skins, employed as a model tissue, were cut into rectangular shape (2.5 cm x 4 cm) inside which the sample membrane (25 × 25 × 0.1 mm, respectively length, width, and thickness) was positioned in the overlapping area. Then, the lap joint was slightly compressed with paperclips and the device put in phosphate buffer solution (PBS, pH=7.4) at 37°C for 30 min to simulate the wet environment of the human body. After this time, the two final parts of the pig skin were fixed to the clamps of tensile machine. The measurements, performed using a shear rate of 0.5 mm/min and a maximum load of 500 N, were conducted until the two pig skins separated. The lap shear strength was determined from the maximum of the experimental stress-strain curve and expressed as a detachment force. Each sample was measured five times and expressed as the mean and standard deviation.”

  and the obtained results (subsection 2.4):

“The performance adhesive of developed bioactive membranes was studied by a lap shear test using fresh pig skins. In figure 9B the values of adhesive strength, expresses as detachment force, of the CS_GMA_GLY20_0.05, CS_GMA_HCAF_GLY20_0.05 and CS_GMA_GLY20_HCAF_0.05 samples were reported. It was possible to notice as the presence of HCAF enhanced the detachment stress value of the bioactive membranes with respect to the free-catechol one, from 30 to 60%. This confirmed that the catechol moieties of the antioxidant could form covalent bonds with functional groups present in skin components. Furthermore, the tissue adhesion properties of the bioactive membranes increased with HCAF concentration increasing (see data reported in results and discussion section). Indeed, the system containing the imbibed antioxidant showed a higher increase in the detachment stress value than compared to that with covalently bound HCAF (1.8 kPa vs 1.4 kPa). These results were in according with those obtained in other literature studies [42-45].”     

  1. The title “application in wound healing”, but in the manuscript I do not see any data about wound healing.

As mentioned above, this work was aimed at verifying the potential application of the developed systems in wound healing field. However, as suggested by the Reviewer, the title of the manuscript has been changed.

In particular, the old title:

“3,4 Hydroxycinnamic acid-containing antimicrobial dressings based on functionalized chitosan for applications in wound healing” has been changed to:

“Development of antioxidant and antimicrobial membranes based on functionalized and crosslinked chitosan for tissue regeneration”.

  1. The Figure should be improved, for example, Figure 8 the font sizes in the pictures are all different, etc.

The resolution of the figures has been improved and the font size of the figure 8 has been equalized.

  1. In the antioxidant assay, I wonder DPPH can represent free radicals existed in the infected tissue, such as hydrogen peroxide and superoxide?

The antioxidant activity of a material can be evaluated using different model radicals such as 2,2-diphenyl-1-picrylhydrazyl radical (DPPH), 2,2azinobis(3-ethylbenzothiazolin-6-sulfonicoacid) cationic radical (ABTS+), the superoxide anion radical or hydroxyl radical. However, the DPPH assay has proven to be one of the most widely used tests for evaluating antioxidant activity of a material as it is capable of capturing free radicals for different systems including composites. Although such test is unable to mimic the radical scavenging mechanism of antioxidants in biological systems due to the lack of oxygen radicals in the test, its use is justified by the assumption that the antioxidant activity is equal to its electron donating capacity. As reported by F. Shahidi et al., the DPPH assay is the first valid approach for evaluating the effectiveness of a systems to scavenge radicals by electron transfer reaction. Furthermore, the EC50 value, determined with this assay, allows different systems to be compare with each other.

  1. Figure 13 was not indexed in the manuscript, and from Figure 13, how do you see a 50% reduction in bacteria

​The Figure 13 was indexed in the manuscript (see line 563 in the manuscript). In the revised manuscript, this figure has been changed to Figure 14.

However, as reported in the manuscript, the Kirby-Bauer test (the results of which can be seen in figure 13) was used to highlight the migration of the bioactive component into the culture medium from the films, in particular from the CS_GMA_GLY20_HCAF_0.05 sample containing the imbibed antioxidant. In this case the formation of inhibition halo of the bacterial growth has to be observed. However, this phenomenon did not occur. Therefore, another more sensitive test (dynamic contact of the strain broth solution with the film) was performed to verify the antimicrobial activity of the developed membranes, as reported in paragraph 3.13. The procedure involved determining the percentage reduction in bacterial growth by means of the difference between the CFU of the control sample and the corresponding ones of the films. This information was reported in the material and methods section, subsection 3.13.

  1. In the introduction, some references are better to cited and discussed in the manuscript, for example, European Polymer Journal, 2024, 202, 112592; Biomedical Engineering Communications 2023;2(1):2; Biomaterials Research 27(1):73, etc.

As suggested by the Reviewer, the references have been cited and discussed in the manuscript. In particular, in the introduction and results and discussion sections (subsection 2.9):

“In particular, endogenous and exogenous stimuli-response systems based on nanomaterials appear to be very promising for biomedical applications including wound healing, as they can not only release the drug in a controlled manner at the target site but also contribute to counteracting the problem of multi-drug resistant bacterial wound infection [13,14].”

“The growing problem of antibiotic resistance of some pathogens and the appearance of micro-organisms resistant to antibiotics have aimed many studies at the development of antimicrobial polysaccharide systems containing natural products such as antimicro-bial peptides or antioxidants. For example, injectable polysaccharide hydrogels for the controlled release of incorporated nisin were investigated by Flynn et al. [76]. Such hydro-gels, composed of oxidised dextran, alginate functionalised with hydrazine groups and glycol chitosan in different percentage, exhibited antimicrobial activity against S. aureus up to 10 days. In addition, it was found that glycol chitosan exerted a synergistic action with nisin in the bacterial growth inhibition. Lee et al., instead, developed chitosan-based systems conjugated with hydroxycinnamic acids including caffeic acid, ferulic acid, and sinapic acid [77]. The conjugates showed improved antimicrobial activity, in terms of minimum inhibitory concentration (MIC), against two standard methicillin-resistant Staphylococcus aureus (MRSA) and foodborne pathogens compared to that of unmodified chitosan. In a very recent study, it was demonstrated that tannic acid (TA) contained in a multifunctional scaffold composed of Schiff base crosslinked konjac glucoman-nan/chitosan hydrogel can stabilize the system structure, modulate its degradation and act as an active drug exerting antioxidant, antibacterial, and anti-biofilm effects [78].”

Reviewer 2 Report

Comments and Suggestions for Authors

The presented  article  of Clarissa Ciarlantini , Elisabetta Lacolla  et al.  ”3,4 Hydroxycinnamic acid-containing antimicrobial dressings based on functionalized chitosan for applications in  wound healing” refers to actual medicine in particular to dressing based on polymer biocompatible  coatings. The technological aspects of development of such means for topical treatment of burns, chronical ulsters  are of great importance for medicine. The current article can be published after accounting some notes.

1.       Literature introduction must include information about patent state in technology of preparing chitosan topical means for  wound healing.

2.       The data about volume and fractional size distribution of pores in chitosan membrane must be added with full description of methodology of size measurement.

3.       The physico-chemical characteristics of offered novel chitosan dressings  is needed to compare with some golden commertial standard  dressing  as example  cellulose  or hyaluronate dressing. The merits and disadvantages of new composition must be discussed in conclusion.

4.       The figures 5 A,B,C,E,F  of electron scan microscopy  don’t represent the pore structure of membrane and must be corrected.

5.       Biocopatibility of chitosan formulation must be supported in tests on normal cells (fibroblasts et cet). Antibacterial action must be  compatible with nontoxic effects  on skin and deep tissues. One is possible to neglect irritational action of modified chitosan on skin?

6.       What is the chemical stability of synthesized chitosan membranes? How long the  physico-chemical membrane characteristics sustain the initial level at room  storage.

7.       Despite both membranes maintaining good values of elongation  at break, water absorption, the scavenging of ROS  and DPPH radicals the final assessment of dressing for practical medicine depends on the preclinical and clinical results. Are there any experimental results on laboratory model burns or wounds?

After revision the article can be published in Int. J. Mol. Sci

Comments on the Quality of English Language

The presented  article  of Clarissa Ciarlantini , Elisabetta Lacolla  et al.  ”3,4 Hydroxycinnamic acid-containing antimicrobial dressings based on functionalized chitosan for applications in  wound healing” refers to actual medicine in particular to dressing based on polymer biocompatible  coatings. The technological aspects of development of such means for topical treatment of burns, chronical ulsters  are of great importance for medicine. The current article can be published after accounting some notes.

1.       Literature introduction must include information about patent state in technology of preparing chitosan topical means for  wound healing.

2.       The data about volume and fractional size distribution of pores in chitosan membrane must be added with full description of methodology of size measurement.

3.       The physico-chemical characteristics of offered novel chitosan dressings  is needed to compare with some golden commertial standard  dressing  as example  cellulose  or hyaluronate dressing. The merits and disadvantages of new composition must be discussed in conclusion.

4.       The figures 5 A,B,C,E,F  of electron scan microscopy  don’t represent the pore structure of membrane and must be corrected.

5.       Biocopatibility of chitosan formulation must be supported in tests on normal cells (fibroblasts et cet). Antibacterial action must be  compatible with nontoxic effects  on skin and deep tissues. One is possible to neglect irritational action of modified chitosan on skin?

6.       What is the chemical stability of synthesized chitosan membranes? How long the  physico-chemical membrane characteristics sustain the initial level at room  storage.

7.       Despite both membranes maintaining good values of elongation  at break, water absorption, the scavenging of ROS  and DPPH radicals the final assessment of dressing for practical medicine depends on the preclinical and clinical results. Are there any experimental results on laboratory model burns or wounds?

After revision the article can be published in Int. J. Mol. Sci

Author Response

Responses to the Reviewer 2.

We thank the Reviewer for agreeing to evaluate the manuscript. We have given significant consideration to the qualified contributions that were found to be fundamental for improving our work's standards of quality and excellence.

  1. Literature introduction must include information about patent state in technology of preparing chitosan topical means for wound healing.

As suggested by the Reviewer the information about the patent state in the technology of preparing chitosan topical means for wound healing has been introduced. In particular, in the Introduction section, a new sentence has been added (line 82):

“Indeed, as reported by P. Shivakumar et al., from 2010 to 2020, several patents on the application of CS and its derivatives in wound healing have been developed, confirming the high applicability of this polymer [20]. The CS ability to limit bleeding phenomena has also allowed the production of commercial dressings such as ChitoGauze XR pro or HemCon GuardaCare® pro.”

  1. The data about volume and fractional size distribution of pores in chitosan membrane must be added with full description of methodology of size measurement.
  2. The figures 5 A, B, C, E, F of electron scan microscopy don’t represent the pore structure of membrane and must be corrected.

In order to more accurately evaluate the morphology of the membranes, the most interesting samples were observed under SEM at higher magnifications (60000X, maximum value available for the instrumentation). Below are the micrographs obtained of the samples: CS_GMA_GLY20_0.05 (A), CS_GMA_HCAF_GLY20_0.05 (B) and CS_GMA_GLY20_HCAF_0.05 (C).

See images in the pdf file.

From the images it was possible to notice the production of highly wrinkled structures, especially in the case of the CS_GMA_HCAF_GLY20_0.05 matrix. However, the presence of microporous structures could not be observed. This limited the evaluation of the volume and fractional size distribution of pores. For this reason, in order to evaluate the presence of nanoporous structures in the developed systems, all the matrices were subjected to porosity measurement using the liquid displacement method, the results of which were reported and discussed in Water Vapor Transmission Rate (WVTR) and porosity measurements (subsection 2.7).

  1. The physico-chemical characteristics of offered novel chitosan dressings is needed to compare with some golden commercial standard dressing as example cellulose or hyaluronate dressing. The merits and disadvantages of new composition must be discussed in conclusion.

AS suggested by the Reviewer, in the revised manuscript, the physico-chemical features of our systems were compared with those of commercial standard dressings, particularly as regard to mechanical and WVTR properties, as discussed in the subsections 2.4 and 2.7, respectively.

  1. Biocopatibility of chitosan formulation must be supported in tests on normal cells (fibroblasts et cet). Antibacterial action must be compatible with nontoxic effects on skin and deep tissues. One is possible to neglect irritational action of modified chitosan on skin?

We agree with the Reviewer. However, in our recent work (Carbohydrate Polymers, 2024, 327, 121684), scaffolds based on chitosan-alginate cross-linked with calcium chloride and containing covalently bound HCAF, showed a good biocompatibility against human fibroblasts and not harmful to the cell viability. Furthermore, several systems based on chitosan modified with GMA reported in literature have demonstrated no toxicity against fibroblast cells. Taking into consideration these results, it was assumed that also our systems could be considered biocompatible. Therefore, in this work, the preparation and physico-chemical characterization of systems with antimicrobial and antioxidant features were of paramount importance. As the obtained results, particularly for the membrane containing the antioxidant physically bound, were very promising, in a future work further in vitro and in vivo biological characterizations will be performed to better define the potential use of this bioactive system for the treatment of wound healing including irritation tests.

  1. What is the chemical stability of synthesized chitosan membranes? How long the physico-chemical membrane characteristics sustain the initial level at room storage.

All the prepared membranes were stable at room temperature as their physical characteristics did not change after a time interval of 30 days. This was verified through both spectroscopic observations, which showed no changes in the chemical composition of the systems, and measurements of swelling and mechanical properties.

  1. Despite both membranes maintaining good values of elongation at break, water absorption, the scavenging of ROS and DPPH radicals the final assessment of dressing for practical medicine depends on the preclinical and clinical results. Are there any experimental results on laboratory model burns or wounds?

Since we don’t have an animal facility, it was not possible to set up experiments on laboratory model burns or wounds.

Reviewer 3 Report

Comments and Suggestions for Authors

Dear Authors,

Submitted work deals with the topical application of dressing to fasten the wound healing process. A series of experiments were performed to characterize the physical, chemical, and biological properties of the 3,4-dihydroxy cinnamic acid-impregnated chitosan-based dressing materials. However, a few experiments were found missing in the submitted manuscript:

1. Kindly provide the biocompatibility data for your formulation. It is advised to perform a skin irritability test on live tissue. Without a biocompatibility study, no topical formulation can be used.

2. No wound healing study was performed. The in-vitro antimicrobial and antioxidant properties can't be considered as an in-vivo wound-healing study. Hence in-vivo wound healing studies are required to justify the title of the manuscript and make the manuscript scientifically sound.

3. If in-vivo wound healing and skin irritability test can't be done then it is advisable to modify the title and replace the word wound healing with antimicrobial and antioxidant.

Author Response

Responses to the Reviewer 3.

We thank the Reviewer for agreeing to evaluate the manuscript. We have given significant consideration to the qualified contributions that were found to be fundamental for improving our work's standards of quality and excellence.

  1. Kindly provide the biocompatibility data for your formulation. It is advised to perform a skin irritability test on live tissue. Without a biocompatibility study, no topical formulation can be used.

We agree with the Reviewer. However, in our recent work (Carbohydrate Polymers, 2024, 327, 121684), scaffolds based on chitosan-alginate cross-linked with calcium chloride and containing covalently bound HCAF, showed a good biocompatibility against human fibroblasts and not harmful to the cell viability. Furthermore, several systems based on chitosan modified with GMA reported in literature have demonstrated no toxicity against fibroblast cells. Taking into consideration these results, it was assumed that also our systems could be considered biocompatible. Therefore, in this work, the preparation and physico-chemical characterization of systems with antimicrobial and antioxidant features were of paramount importance. As the obtained results, particularly for the membrane containing the antioxidant physically bound, were very promising, in a future work further in vitro and in vivo biological characterizations will be performed to better define the potential use of this bioactive system for the treatment of wound healing including irritation tests.

  1. No wound healing study was performed. The in-vitro antimicrobial and antioxidant properties can't be considered as an in-vivo wound-healing study. Hence in-vivo wound healing studies are required to justify the title of the manuscript and make the manuscript scientifically sound.

We agree with the Reviewer. However, we don’t have an animal facility in Department. Therefore, it was not possible to set up experiments in-vivo wound-healing study. As suggested by the Reviewer, in the next comment, the title of manuscript “3,4 Hydroxycinnamic acid-containing antimicrobial dressings based on functionalized chitosan for applications in wound healing” has been changed to:

 “Development of antioxidant and antimicrobial membranes based on functionalized and crosslinked chitosan for tissue regeneration”.

  1. If in-vivo wound healing and skin irritability test can't be done then it is advisable to modify the title and replace the word wound healing with antimicrobial and antioxidant.

As stated above, the title has been changed.

Reviewer 4 Report

Comments and Suggestions for Authors

1. please describe in more detail what the actual mechanism of crosslinking is also, why is SMBS added to PPS and the role of HCAF?

2. are these materials likely to be biocompatible we have no results to back this up?

3. no control of antimicrobial activity and results are unclear, please compare with results reported here for similar materials Journal of Materials Chemistry B 8 (18), 4029-4038

4. why is 5D so different is it swollen by solvent during the crosslinking procedure

5 the derivative of plots 6 and 7 would be useful

6. please report stiffness values.

7. please report contact angle images

8. 

Comments on the Quality of English Language

minor check required

Author Response

Responses to the Reviewer 4.

We thank the Reviewer for agreeing to evaluate the manuscript. We have given significant consideration to the qualified contributions that were found to be fundamental for improving our work's standards of quality and excellence.

  1. please describe in more detail what the actual mechanism of crosslinking is also, why is SMBS added to PPS and the role of HCAF?

Generally, in the case of radical polymerization in an aqueous environment, sodium metabisulfite (SMBS) and potassium persulfate (PPS) are used as initiators to produce radicals necessary to activate the double bond of appropriate monomers. In this work, SMBS and PPS were used to produce radicals necessary to activate the glycidyl methacrylate monomer (GMA) and ethylene glycol dimethacrylate (EGDMA), cross-linking agent able to linking GMA-functionalized CS chains together. Such initiators allowed to carry out the reaction at room temperature for short times avoiding degradation phenomena that the system could undergo.

HCAF (3,4 dihydroxycinnamic acid), as shown by antioxidant tests, is a molecule with high radical capture capabilities. For this reason, phenomena of capture of radicals produced in the cross-linking phase by HCAF could not be excluded. However, the change in the mechanical and swelling properties of the developed systems after HCAF introduction suggested that this phenomenon might be negligible. In fact, for both methodologies used to introduce the antioxidant molecule, it was possible to notice a decrease in the swelling capacity and an increase in the elastic modulus value, confirming the cross-linking process had taken place.

  1. are these materials likely to be biocompatible we have no results to back this up?

In our recent work (Carbohydrate Polymers, 2024, 327, 121684), scaffolds based on chitosan-alginate cross-linked with calcium chloride and containing covalently bound HCAF, showed a good biocompatibility against human fibroblasts and not harmful to the cell viability. Furthermore, several systems based on chitosan modified with GMA reported in literature have demonstrated no toxicity against fibroblast cells. Taking into consideration these results, it was assumed that also our systems could be considered biocompatible. Therefore, in this work, the preparation and physico-chemical characterization of systems with antimicrobial and antioxidant features were of paramount importance. As the obtained results, particularly for the membrane containing the antioxidant physically bound, were very promising, in a future work further in vitro and in vivo biological characterizations will be performed to better define the potential use of this bioactive system for the treatment of wound healing including irritation tests.

  1. no control of antimicrobial activity and results are unclear, please compare with results reported here for similar materials Journal of Materials Chemistry B 8 (18), 4029-4038

As suggested by the Reviewer, the results obtained in this work regarding our bioactive membranes have been compared to similar materials reported in literature (see subsection 2.9).

  1. why is 5D so different is it swollen by solvent during the crosslinking procedure

The CS_GMA_GLY20_0.5 sample swelled less than the others. The rough surface structure of this sample was due to the high concentration of EGDMA used in the crosslinking reaction (0.5 mM). In fact, only in this case, the matrix showed the formation of surface roughness, evident from the SEM micrograph, and a reduction of the film dimension (shrinkage of the film).

  1. the derivative of plots 6 and 7 would be useful

To determine the decomposition stages of the developed systems and the degradation temperature values (Td °C), these latter reported in Table 1, the second derivative of the curves was used. Since the systems developed were numerous as well as the figures reported in the manuscript, it was decided to report only the thermogravimetric curves. However, as suggested by the Reviewer, the derivatives of thermal plots of the figure 6 and 7 have been reported in the revised manuscript.

  1. please report stiffness values.

Stiffness represents resistance to deformation. In particular, stiffness value is determined by taking the slope of the elastic section of the stress-strain curve of a given material. The main property that expresses stiffness of a material is the elastic modulus value. The elastic modulus values measured for our systems were reported in Table 1.

  1. please report contact angle images

As suggested by the Reviewer, images of static contact angle measurements of some selected samples for which discrete differences in contact angle values were found have been included in the manuscript (see figure 12 in subsection 2.6).

Round 2

Reviewer 2 Report

Comments and Suggestions for Authors

The authors did a lot of work to improve the quality of content article. Article name is changed  in favor of correspondence to general item of tissue regeneration means. The new offered name “Development of antioxidant and antimicrobial membranes based on functionalized and crosslinked chitosan for tissue regeneration” better reflects the task and achieved results. The reference [20] gives the indicative to patent applications in chitosan dressings for wound healing. The authors took into account the note about  the need to compare the characteristics of  modified chitosan membrane  with current commercial polysaccharide-based dressings as as Aquacel ® Ag Hydrofiber dressing, composed of sodium carboxymethylcellulose (CMC) with silver ions incorporated, or  Kaltostat ® dressing, based on alginate fibers  410-419. The added porosity data performed with the 524 liquid displacement method confirmed  the large porosity for CS and CS_GMA membranes. The methology of porosity measurements was described in 3.11 Materials and methods.

In final conclusions the authors  admit the critical remark about “however, further in vitro 872 and in vivo biological characterizations will be required to better define the potential use 873 of this bioactive system for the treatment of wound healing”

In my opinion the  made corrections and additions  meet critical requirments and article can be submitted for publication in  Int.J.Mol.Sci.

Comments on the Quality of English Language

no comments

Reviewer 3 Report

Comments and Suggestions for Authors

Dear Authors,

Cytotoxicity study against cell lines can not be treated as a skin compatibility test. It is a routine protocol to do skin irritability and anti-inflammatory tests for the topical formulations. We can't assume biocompatibility. However, with the change of title and inclusion of reference number 20 (Shivakumar, P.; Gupta, M.S.; Jayakumar, R.; Gowda, D.V. Prospection of Chitosan and Its Derivatives in Wound Healing: Proof 925 of Patent Analysis (2010–2020). Int. J. Biol. Macromol. 2021, 184, 701–712, doi:10.1016/j.ijbiomac.2021.06.086.) this issue becomes irrelevant to this manuscript. Further, the Non-availability of Animal houses can't be a justification for the experiments not being performed.